# Extracellular bacterial lymphatic metastasis drives *Streptococcus pyogenes* systemic infection

Matthew K. Siggins [1,2,5✉], Nicola N. Lynskey[1,2,6], Lucy E. Lamb[1], Louise A. Johnson[3], Kristin K. Huse[1,2], Max Pearson [1,2], Suneale Banerji[3], Claire E. Turner[1,7], Kevin Woollard [4], David G. Jackson [3] & Shiranee Sriskandan [1,2✉]

Unassisted metastasis through the lymphatic system is a mechanism of dissemination thus far ascribed only to cancer cells. Here, we report that *Streptococcus pyogenes* also hijack lymphatic vessels to escape a local infection site, transiting through sequential lymph nodes and efferent lymphatic vessels to enter the bloodstream. Contrasting with previously reported mechanisms of intracellular pathogen carriage by phagocytes, we show *S. pyogenes* remain extracellular during transit, first in afferent and then efferent lymphatics that carry the bacteria through successive draining lymph nodes. We identify streptococcal virulence mechanisms important for bacterial lymphatic dissemination and show that metastatic streptococci within infected lymph nodes resist and subvert clearance by phagocytes, enabling replication that can seed intense bloodstream infection. The findings establish the lymphatic system as both a survival niche and conduit to the bloodstream for *S. pyogenes*, explaining the phenomenon of occult bacteraemia. This work provides new perspectives in streptococcal pathogenesis with implications for immunity.

[1] Department of Infectious Disease, Imperial College London, London W12 0NN, UK. [2] MRC Centre for Molecular Bacteriology and Infection, Imperial College London, London SW7 2DD, UK. [3] MRC Human Immunology Unit, MRC Weatherall Institute of Molecular Medicine, University of Oxford, Oxford OX3 9DS, UK. [4] Centre for Inflammatory Disease, Department of Immunology & Inflammation, Imperial College London, London W12 0NN, UK. [5] Present address: NLHI, Imperial College London, London W2 1PG, UK. [6] Present address: The Roslin Institute, University of Edinburgh, Edinburgh EH25 9RG, UK. [7] Present address: The Florey Institute, University of Sheffield, Sheffield S10 2TN, UK. ✉email: m.siggins@imperial.ac.uk; s.sriskandan@imperial.ac.uk

Lymphatic vessels form an extensive, tissue-permeating network around the body that carries fluid from interstitial spaces, through lymph nodes, and back to the blood circulation. Initial lymphatics have distinctive loose junctions that allow selective passage of leucocytes, fluids and small macromolecules but could in principle also provide an ideal conduit for the dissemination of pathogens[1,2]. However, in addition to their well-recognised importance in adaptive immunity, lymph nodes are reported to efficiently filter microbes[3,4] and orchestrate innate immune responses that contain and kill pathogens to limit systemic dissemination[5]. Hence, lymphatic metastasis—the spread of a pathogenic agent from an initial to a secondary site via the lymphatics—is only considered to contribute meaningfully to the pathogenesis of some cancers[6,7] and a few specialised intracellular lymph-tropic pathogens, such as Yersinia pestis which can be carried through the lymphatics inside phagocytic cells[8]. The human pathogen Streptococcus pyogenes (group A streptococcus) is widely recognised to interact with the lymphatic system presenting clinical manifestations such as lymphadenitis and lymphangitis[9–11]. We recently provided a possible explanation for this tropism, by demonstrating that the hyaluronan capsule of S. pyogenes binds LYVE-1 (lymphatic vessel endothelial receptor), the hyaluronan receptor that mediates leukocyte entry to initial lymphatic vessels[12,13].

S. pyogenes is a major infectious cause of global human mortality[14] and the leading bacterial cause of tonsillitis and cellulitis. Invasive streptococcal infection, albeit rare, is particularly devastating, with case-fatality rates of up to 25%[14,15]. One fifth of invasive S. pyogenes infections present as bloodstream infection without any obvious point of origin[16,17], and almost half of invasive streptococcal necrotising soft tissue infections lack any clear portal of entry[18–20]. It has been postulated that S. pyogenes emanates from a mucosal site of colonisation or infection to seed soft tissue infection via the bloodstream, although this has been neither demonstrated nor the mechanism of entry to the bloodstream elucidated[21,22]. We hypothesised that S. pyogenes tropism for the lymphatic system is important in such occult infections.

Here we show that virulent S. pyogenes not only reach the first local draining lymph node, as demonstrated previously[12], but readily transit through sequential lymph nodes within efferent lymphatics to reach the bloodstream, while remaining extracellular. This lymphatic metastasis is a key pathway of dissemination that can drive bloodstream infection.

## Results

**S. pyogenes access systemic circulation via lymphatics.** Based on the lymphatic pathology associated with S. pyogenes disease and our recent discovery that the hyaluronan capsule of S. pyogenes binds the lymphatic endothelial hyaluronan receptor LYVE-1[12,23], we hypothesised that the lymphatic system plays a central role in streptococcal pathogenesis in invasive infection. To investigate this, we first systematically studied lymphatic involvement in bacterial dissemination using an established murine model of S. pyogenes soft tissue invasion[24]. Following intramuscular infection, we quantified bacterial burden in systemic organs, all lymph nodes that drain the hindlimb, and nodes that do not drain this injection site (Supplementary Fig. 1). Within 0.5 h, mice infected with S. pyogenes invasive M1T1 clinical isolate H598 (Supplementary Table 1) had significant numbers of viable S. pyogenes in not only the local draining inguinal lymph node but also the sequential distant-draining ipsilateral axillary lymph node. In contrast, no bacteria were in any non-draining lymph nodes, and few bacteria were recovered from blood or systemic organs (Fig. 1a). These observations pointed to a fundamentally distinct route of dissemination for extracellular bacteria, involving

not only transit via draining afferent lymphatics to a local draining lymph node, but also subsequent transit through efferent postnodal lymphatics in order to reach sequential distant draining lymph nodes. 3 h after infection, bacterial numbers in local and distant draining lymph nodes had increased 25-fold, whereas non-draining lymph nodes remained clear of bacteria (Fig. 1b). Draining lymph finally enters the bloodstream via the lymphatic ducts, and the surge in lymphatic bacterial burden at 3 h was accompanied by a similar surge in spleen bacterial load, an organ which removes bacteria from blood, providing clear evidence of S. pyogenes entry into the bloodstream[25]. 24 h post-infection, bacteraemia had increased from <10 CFU/ml to ≥10^6 CFU/ml and a high bacterial burden was identified in all organs (Fig. 1c, d) including both draining and non-draining lymph nodes, consistent with blood-borne systemic spread. Significantly, bacterial counts within the infection site in the hindlimb remained constant throughout the infection time course, suggesting that the lymphatic dissemination and metastatic expansion of bacteria was driving development of intense bloodstream infection.

To determine if transit through the draining inguinal lymph node was required for bacteria to reach the sequential distant draining axillary node, the local draining inguinal lymph node was excised prior to hindlimb infection. This lymphadenectomy abrogated bacterial dissemination to the axillary lymph node (Fig. 1e), providing clear confirmation that transit of bacteria from the infection site to distant lymph nodes occurred via sequential lymph nodes (Fig. 1f). Importantly, a range of different S. pyogenes clinical isolates exhibited bacterial dissemination via the lymphatic system across different phases of bacterial growth and within FVB/n, BALB/c, and C57BL/6 mouse strains (Supplementary Fig. 2a, b), implying broad significance to streptococcal infection. Notably, lymphatic dissemination occurred with a range of infection titres (Supplementary Fig. 2c), including bacterial densities similar to those encountered in human tissues during S. pyogenes infection[26], underlining the relevance of our findings to human disease.

**S. pyogenes transit to and between lymph nodes in lymphatic vessels.** Next, to unequivocally confirm the route of bacterial transit between lymph nodes, we directly monitored the efferent vasculature that runs from the inguinal to the axillary lymph node using intravital confocal microscopy and a hypervirulent in vivo-passaged M1T1 strain (H1565) that displayed prominent lymphatic dissemination (Fig. 2a–c and Supplementary Fig. 3a–c and Supplementary Movie 1). Within 1 h of injection into the hindlimb, fluorescent S. pyogenes (Fig. 2d) were present inside postnodal efferent lymphatic vessels that transport lymph from the inguinal to the axillary node but were absent from blood vessels and surrounding tissue (Fig. 2e). Intraluminal bacteria appeared to be either suspended in lymph flow or bound to lymphatic endothelial surfaces (Fig. 2f, g and Supplementary Movies 2 and 3). In some lymphatic vessels, we observed streptococci in close association with leucocytes, although careful analysis revealed these were not intracellular but rather attached to the leucocyte cell surface in characteristic small chains (Fig. 2h, i, and Supplementary Movies 4 and 5). Moreover, phenotypic analyses indicated most leucocytes within the efferent lymphatic vessel were CD3^+ T cells with only small numbers of CD11b^+ phagocytes being present (Supplementary Data Fig. 3h, i).

Tumour cells, which also can transit through sequential lymph nodes to reach the blood[27], have recently been reported to intravasate lymph node blood vessels to access the systemic circulation directly from within lymph nodes[28,29]. To ascertain whether bacteria were invading lymph node blood vessels to reach the systemic circulation or remained predominantly within

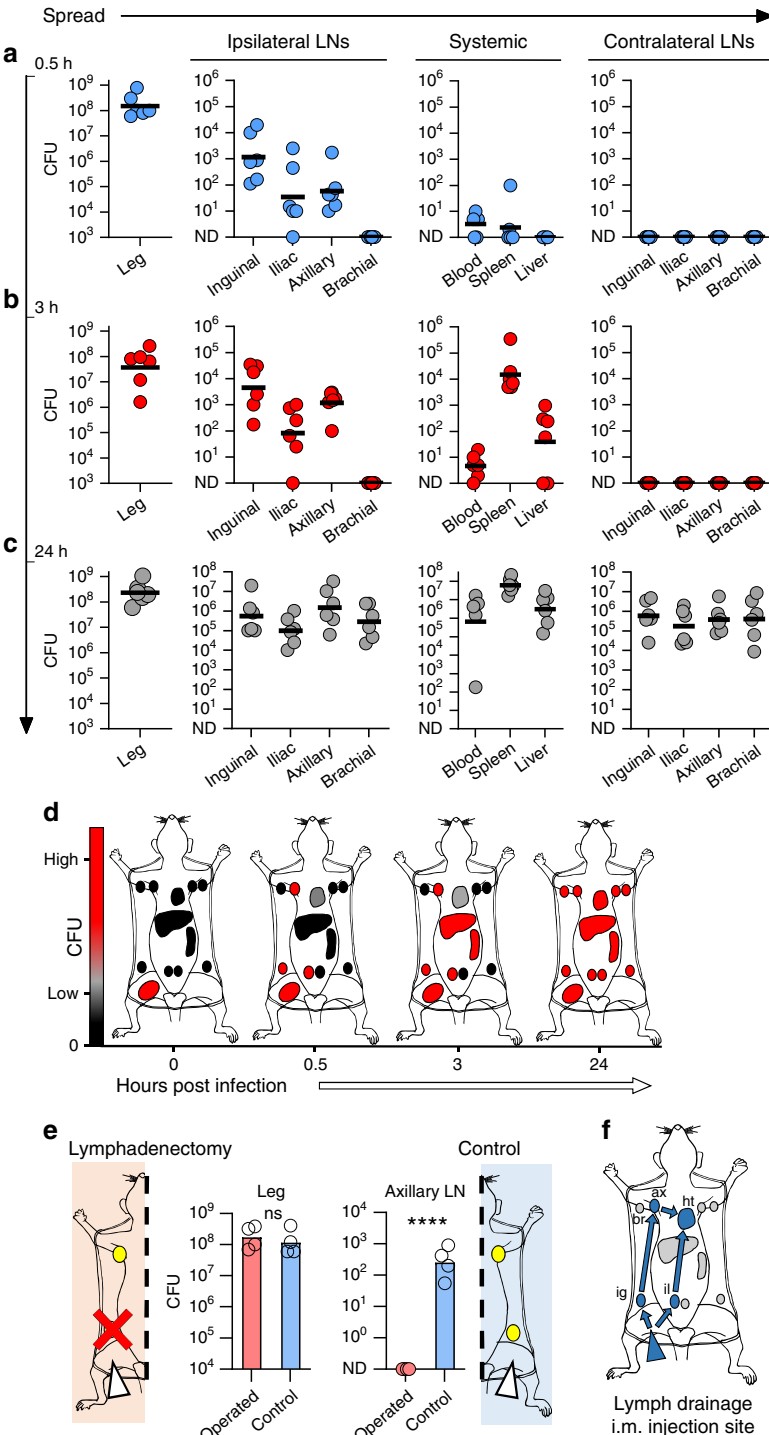

**Fig. 1 S. pyogenes access systemic circulation via lymphatics. a–c** S. pyogenes (H598) recovered from the hindlimb infection site, lymph nodes, and systemic organs of FVB/n mice 0.5 h (blue circles, **a**), 3 h (red circles, **b**), or 24 h (grey circles, **c**) after intramuscular infection with $10^8$ CFU. Symbols represent individual mice, $n = 6$ per group, lines indicate geometric means. **d** Schematic summarising representative time course of transit of S. pyogenes from the hindlimb following intramuscular injection of hypervirulent S. pyogenes, organs listed in f. Black indicates 0 CFU; grey, low CFU; and red, high CFU counts. **e** Recovery of S. pyogenes from hindlimb infection site and ipsilateral axillary lymph nodes 1 h after intramuscular infection with $10^8$ CFU in FVB/n mice, with (red bars or red circles) or without (blue bars) surgical removal of the draining ipsilateral inguinal lymph node prior to infection. Symbols represent individual mice, $n = 4$ per group, bars indicate geometric means. ****$p \leq 0.0001$; ns, $p > 0.5$: Two-tailed Student's t-test performed on log10-transformed data. **f** Schematic of lymphatic drainage route from hindlimb following intramuscular injection of Evans Blue dye, indicated by blue arrows and colouring: ig, ipsilateral inguinal node; il, ipsilateral iliac node; ax, ipsilateral axillary node; and ht, heart (via subclavian vein); br, ipsilateral (non-draining) brachial node. See also Supplementary Figs. 1 and 2. CFU are per ml of blood, per g of liver, per leg, or per organ.

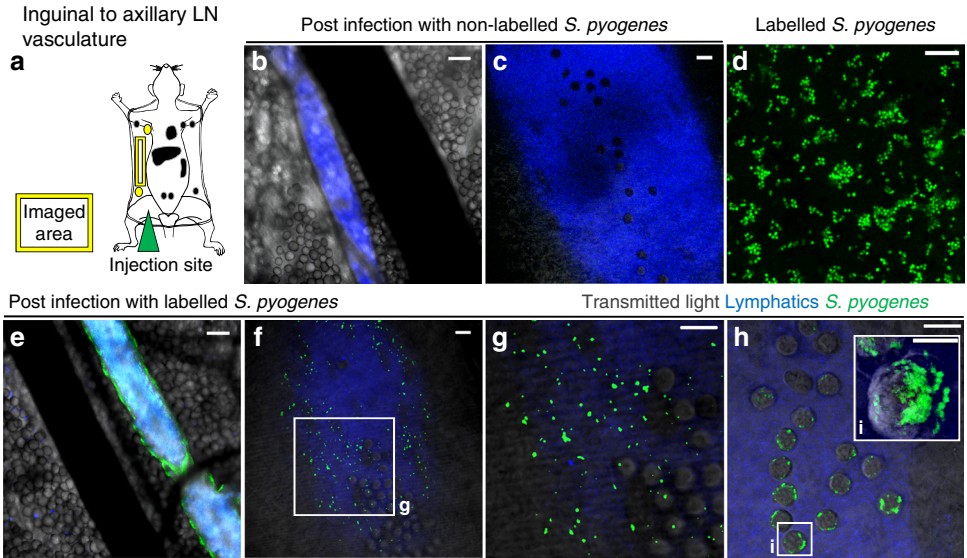

**Fig. 2 *S. pyogenes* transit to and between lymph nodes in lymphatic vessels. a** Schematic of imaging area. **b–h** Intravital microscopy images of the efferent lymphatic vessel that connects the inguinal and axillary lymph nodes in mice. *S. pyogenes* (green), lymph (blue) and blood vessels (black). FVB/n mice infected intramuscularly with $10^8$ CFU of hypervirulent *S. pyogenes* (H1565), either unlabelled (**b**, **c**) or fluorescently labelled (**e–h**) and imaged within 1 h. Scale bars: 100 μm (**b**, **e**); 10 μm (**c**, **d**, **f–i**). **i** Three-dimensional reconstruction of fluorescently labelled streptococci adhered to leukocyte cell surface, scale bar 5 μm. Data are representative of four independent experiments. See also Supplementary Fig. 3 and Supplementary Movies 1–5.

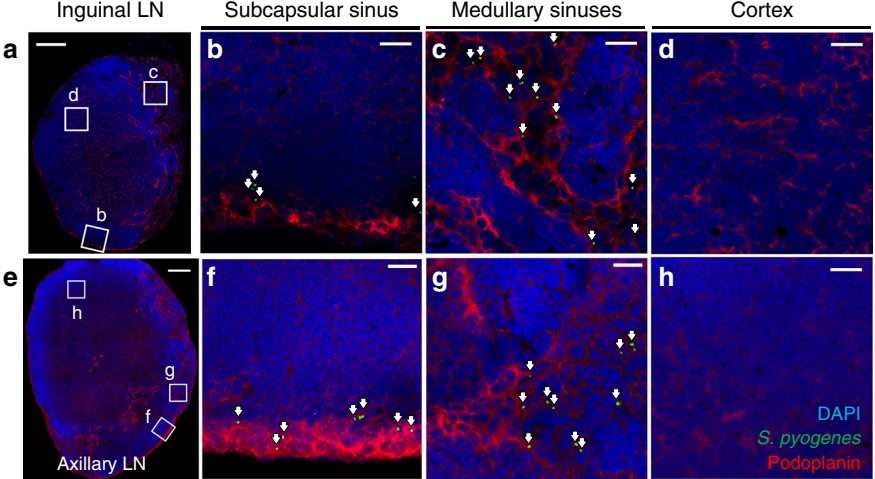

**Fig. 3 *S. pyogenes* are confined to sinuses within lymph nodes. a–h** Immunofluorescence staining of local draining inguinal (**a–d**) and distant draining axillary (**e–h**) lymph node cryosections from FVB/n mice prepared 3 h after intramuscular injection into the hindlimb with $10^8$ CFU of hypervirulent *S. pyogenes* (H1565); stained for *S. pyogenes* (green), podoplanin (red) and with DAPI (blue). White arrows highlight the location of *S. pyogenes*. Podoplanin staining highlights lymphatic endothelial cells in the subcapsular and medullary sinuses, as well as fibroblastic reticular cells in the cortex of the lymph node. Scale bars: 200 μm (**a**, **e**); 20 μm (**b–d**, **f–h**). Data are representative of five independent experiments. See also Supplementary Fig. 4a, b, and Supplementary Movies 6 and 7.

the lymph node sinuses, we assessed the position of *S. pyogenes* within draining lymph nodes 3 h after infection with the hypervirulent strain H1565 using immunofluorescence microscopy. In both ipsilateral inguinal lymph nodes (local draining) (Fig. 3a–d and Supplementary Movie 6) and ipsilateral axillary lymph nodes (distant draining) (Fig. 3e–h and Supplementary Movie 7), bacteria accumulated in the subcapsular (Fig. 3b, f) and medullary sinuses (Fig. 3c, g), consistent with entry and exit of bacteria between sequential lymph nodes via the lymphatics. In contrast, very few streptococci were observed to have exited to the lymph node parenchyma (Fig. 3d, h), or to have associated with high endothelial venules, indicating a lack of direct bacterial entry or exit through blood vessels in early infection. Even 24 h after

infection, streptococci in ipsilateral axillary lymph nodes were predominantly seen in subcapsular sinuses, although an association with high endothelial venules was evident at this time point within non-draining lymph nodes, indicative of bloodstream spread (Supplementary Fig. 4a, b).

**Bacterial virulence factors determine extent and consequences of lymphatic-dissemination.** In principle, the small dimensions of bacteria alone might be sufficient to permit passive dissemination via lymphatic channels. To determine whether the process depended on the active involvement of bacterial virulence mechanisms, we evaluated a panel of isogenic *S. pyogenes* strains.

Invasive streptococcal isolates that cause severe systemic infections, such as those used in our study, frequently carry mutations in the bacterial two-component regulatory system CovR/S (also known as CsrRS)[30–32], which de-repress virulence factors that promote invasiveness, including hyaluronan capsule and the chemokine-cleaving protease, SpyCEP[33]. We compared the in vivo-passaged hypervirulent strain H1565, which has a CovR mutation and therefore produces abundant capsule (high-capsule CovR[mutant]), with its parent clinical M1T1 strain H584 (wildtype) in a 24-h time course infection. We further compared these strains with additional isogenic derivatives of H584 that either lacked capsule (Δcapsule, H1454), or produced abundant capsule (high-capsule ΔP2, H1458), due to a specific mutation in the capsule synthesis promoter[34] (Supplementary Table 1). While bacterial counts at the injection site were similar for all strains throughout infection, and all strains exhibited some lymphatic dissemination, the two high-capsule strains exhibited a particular propensity for lymphatic dissemination and also had greater bacterial loads in systemic organs during the first 3 h of infection than wildtype and acapsular strains (Fig. 4a, b), confirming a specific role for S. pyogenes hyaluronan capsule in bacterial lymphatic tropism and retention. Abrogating LYVE-1-mediated retention of hyaluronan-expressing S. pyogenes within the lymphatic system[12], using a LYVE-1 hyaluronan blocking antibody[23], significantly reduced bacterial numbers in both the local draining inguinal and distant draining axillary lymph nodes, but simultaneously augmented the entry of bacteria into the bloodstream, spleen and liver (Supplementary Fig. 4c–e). Hence, functional blockade of LYVE-1 allows S. pyogenes that reach the draining lymph node to more readily transit along lymphatic vessels, through lymph nodes and into the bloodstream. By 24 h postinjection, infections with the high-capsule CovR[mutant] strain had progressed to a much higher systemic bacterial load than high-capsule ΔP2 strain and featured more prominent seeding of nondraining lymph nodes than all the other strains (Fig. 4c). These data pointed to a role for further CovR/S-regulated virulence factors, in addition to the hyaluronan capsule, that enable the subsequent expansion of bacterial numbers within lymph node infection metastases and drive development of systemic disease.

**S. pyogenes subvert recruitment of neutrophils to aid survival within lymph nodes.** Neutrophils are rapidly recruited to infected tissues and the local draining lymph node in response to an assortment of chemoattractants produced by various cells in inflamed tissues[5,35,36]. Having identified a key role for CovR/S-mediated virulence in the expansion of S. pyogenes within lymph node metastases, we hypothesised that the CovR/S-regulated chemokine-cleaving protease, SpyCEP, which can inhibit neutrophil-recruitment to an infection site[24,37] might play a role

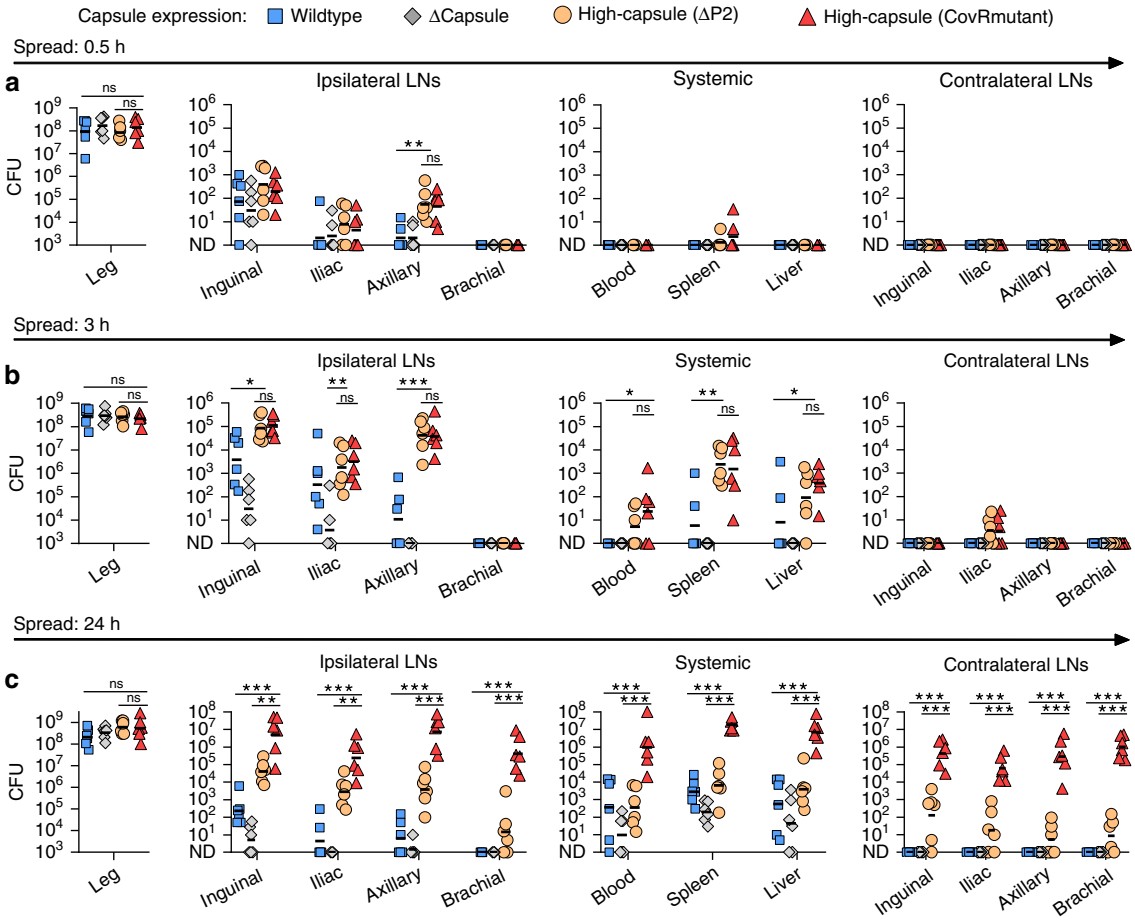

**Fig. 4 Bacterial virulence factors determine the extent and consequences of lymphatic-dissemination. a–c** Recovery of a panel of isogenic S. pyogenes isolates from the hindlimb infection site, lymph nodes, and systemic organs of FVB/n mice at 0.5 h (**a**), 3 h (**b**), or 24 h (**c**) after intramuscular infection with 10[8] CFU. Isogenic strains were: wildtype (H584); acapsular mutant, Δcapsule (H1454); high-capsule ΔP2 (H1458); and high-capsule CovR[mutant] (H1565). Symbols represent individual mice, n = 6 per group, black lines indicate geometric means. *p ≤ 0.05; **p ≤ 0.005; ***p ≤ 0.001; ns, p > 0.5: one-way ANOVA followed by Tukey's multiple comparisons post-tests were performed on log10-transformed data. CFU are per ml of blood, per g of liver, per leg, or per organ.

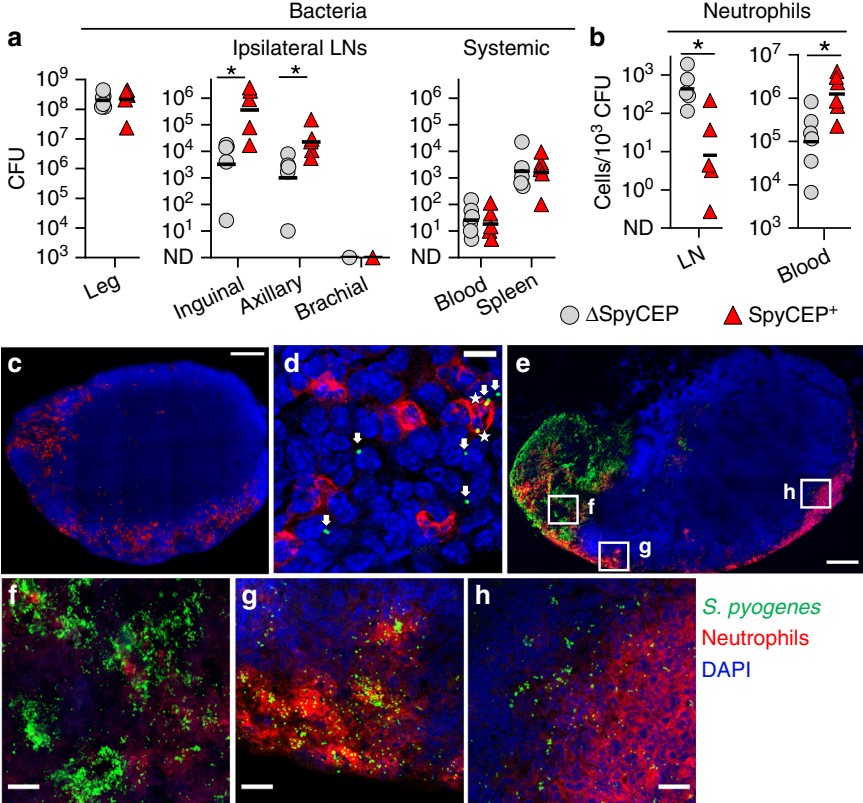

**Fig. 5 S. pyogenes subvert recruitment of neutrophils to aid survival within lymph nodes. a, b** Recovery of bacteria (**a**) and neutrophil counts (CD45+, CD11b+, Ly6G+, live singlets) relative to bacterial load (**b**), 3 h after hindlimb intramuscular infection of FVB/n mice with $10^8$ CFU of SpyCEP-deficient *S. pyogenes* (grey circles, H1567) or an isogenic SpyCEP+ strain (red triangles, H1565). Symbols represent individual mice and black lines indicate geometric means. *n* = 6 (Leg and Blood) or *n* = 5 (LNs and Spleen) per group. *$p \leq 0.05$; inguinal, $p = 0.0180$; axillary, $p = 0.0440$ (**a**); LN, $p = 0.0117$; blood, $p = 0.0120$ **b** two-tailed Student's *t*-test performed on $\log_{10}$-transformed data. **c–h** Immunofluorescence images of cryosections from a draining inguinal lymph node 3 h (**c, d**) or 24 h (**e–h**) after intramuscular infection of FVB/n mice with $10^8$ CFU of *S. pyogenes* H1565 (green); neutrophils (red), and nuclei (blue). Extracellular and intracellular streptococci indicated by arrows and asterisks, respectively (**d**). Scale bars represent 200 μm (**c, e**) and 20 μm (**b, f–h**). Imaging data are representative of five independent experiments. See also Supplementary Fig. 5 and Supplementary Movie 9. CFU are per ml of blood, per leg, or per organ.

in survival and expansion of *S. pyogenes* within lymph nodes. To determine if SpyCEP-mediated subversion of neutrophil-recruitment might underpin the ability of hypervirulent *S. pyogenes* to spread within the lymphatic system, we generated a ΔSpyCEP mutant (H1567) from the hypervirulent strain H1565 and measured numbers of bacteria and neutrophils in lymph nodes following hindlimb infection. Compared to its parent strain H1565, infection with the ΔSpyCEP mutant yielded a 100-fold and 20-fold reduction in bacterial counts in draining ipsilateral inguinal and sequential axillary lymph nodes respectively, despite no significant change in CFU at the infection site at the early 3 h time point (Fig. 5a). Infection with the ΔSpyCEP mutant also resulted in 50-fold higher relative neutrophil numbers (live CD45+, CD11b+, Ly6G+ cells) in draining lymph nodes compared to the virulent strain and 10-fold lower neutrophil counts in blood (Fig. 5b), consistent with a major role for the chemokine CXCL8 (CXCL1/2 in mice) in neutrophil recruitment to the lymph node[36]. Additionally, fluorescent immunostaining revealed only limited phagocytosis of hypervirulent *S. pyogenes* H1565 by neutrophils in lymph nodes (Fig. 5c, d). By 24 h post-infection, extensive prominent foci of invading bacteria were evident deeper in the lymph node parenchyma, often forming large extended necrotic regions (Fig. 5e and Supplementary Movie 8). In these areas, diffuse nucleic acid staining demonstrated significant lysis of cells, including neutrophils (Fig. 5f). Where *S. pyogenes* had

expanded into deeper sinuses further into the parenchyma of draining lymph nodes, there was a relative paucity of recruited neutrophils in comparison with the subcapsular sinus (Fig. 5g, h), suggesting that invading bacteria may be partly sheltered from the immune response due to the architecture of the lymph node impeding cellular recruitment. The hypervirulent *S. pyogenes* strain used in these experiments is inherently more resistant to phagocytosis by neutrophils than wild type *S. pyogenes*, expressing factors other than SpyCEP that impair phagocytic clearance, including capsule (Supplementary Fig 5a). Both SpyCEP+ and ΔSpyCEP *S. pyogenes* were able to multiply in lymph nodes over 24 h despite the ratio of bacteria to neutrophils remaining visibly higher in mice infected with the SpyCEP+ strain (Supplementary Fig. 5b–g). Confocal microscopy revealed that most of these bacteria were extracellular (Supplementary Fig. 5h). However, isolation of the role of SpyCEP through heterologous expression of SpyCEP in *Lactococcus lactis* (an avirulent bacterium that is readily phagocytosed), revealed that SpyCEP drove systemic infection in mice, dramatically increasing bacterial burden in draining lymph nodes and systemic organs (Supplementary Fig. 5i). We conclude that the CovR/S-regulated virulence factor SpyCEP compromises recruitment of neutrophils to help drive progression of infection in draining lymph nodes, although additional CovR/S factors also promote bacterial replication and survival.

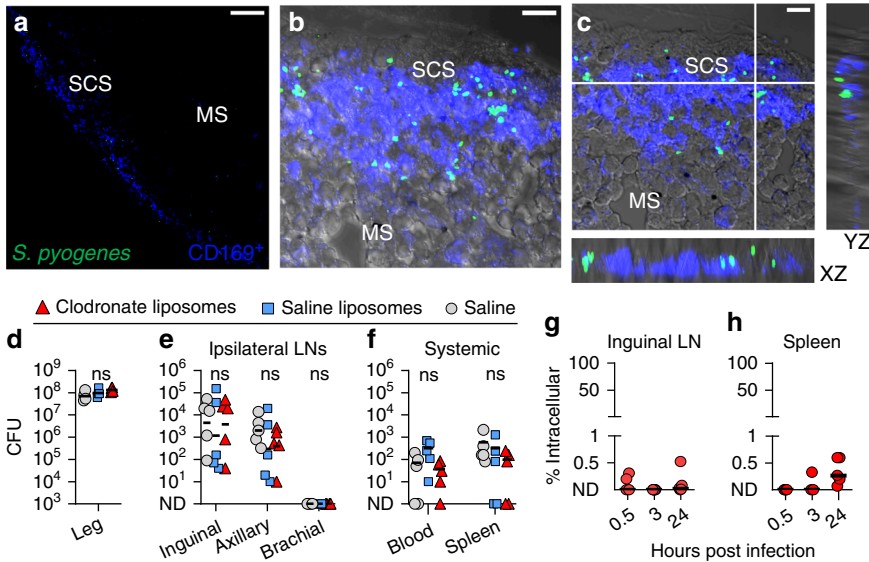

**Fig. 6 S. pyogenes is extracellular within lymph nodes and evades killing by CD169[+] macrophages. a–c** Immunofluorescence staining of cryosections from the draining inguinal lymph nodes of FVB/n mice 3 h after intramuscular injection into the hindlimb with 10[8] CFU of hypervirulent S. pyogenes H1565 (green); CD169 (blue) and brightfield shown in grey; Overview of the subcapsular sinus (SCS) and medullary sinuses (MS) (**a**); Maximum intensity projection (**b**) and orthogonal views of a 15 μm confocal z-stack (**c**). Scale bars: 50 μm (**a**); 10 μm (**b**, **c**). Imaging data are representative of five independent experiments. See also Supplementary Movie 8. **d–f** Recovery of S. pyogenes H1565 from the hindlimb infection site (**d**), draining lymph nodes (**e**), and systemic organs (**f**) of FVB/n mice injected subcutaneously in the tail and leg with either clodronate liposomes (red triangles), saline liposomes (blue squares), or a saline sham (grey circles) 96 h prior to a 3-h intramuscular infection with 10[8] CFU. Symbols represent individual mice, n = 5 per group, lines indicate geometric means. ns, p > 0.5: one-way ANOVA followed by Tukey's multiple comparisons post-tests were performed on log₁₀-transformed data. **g**, **h** Percentage of intracellular S. pyogenes 0.5, 3, or 24 h after intramuscular infection with 10[8] CFU of S. pyogenes H1565; gentamicin protection assays on ipsilateral inguinal nodes (**g**) or spleens (**h**) of infected FVB/n mice at each time point. Each red circle represents an individual mouse, n = 5 per group, black lines indicate geometric means. See also Supplementary Fig. 6. CFU are per ml of blood, per leg, or per organ.

**S. pyogenes is extracellular within lymph nodes and evades killing by CD169[+] macrophages.** The lymph node sinuses, where we observed metastatic S. pyogenes to accumulate during infection, are lined with CD169[+] macrophages, including subsets well-recognised for capture of certain viruses and restriction of viral systemic dissemination[38–40]. However, our data showed that many streptococci transit through sequential lymph nodes, apparently unhindered by this population of macrophages. We investigated this interaction using confocal microscopy, which showed that 3 h after infection with a hypervirulent strain (H1565), S. pyogenes were neither sequestered nor internalised by CD169[+] cells, with only extremely rare instances of colocalization (Fig. 6a–c, Supplementary Movie 9). Furthermore, depletion of all lymph-borne macrophages with clodronate liposomes (Supplementary Fig 6a, b) prior to infection with S. pyogenes had no effect on either lymph node or systemic bacterial counts (Fig. 6d–f). Gentamicin protection assays performed on single-cell suspensions of lymph nodes (Fig. 6g, h) also showed that bacteria within lymph nodes were overwhelmingly extracellular throughout the infection time course. In co-culture, M1T1 wildtype H598 and otherwise isogenic highly encapsulated ΔP2 and CovR^mutant strains showed only limited internalisation by murine bone marrow-derived macrophages (Supplementary Fig. 6c, d). Moreover, these bacteria all induced significant macrophage cell death (Supplementary Fig. 6e), supporting the observation that lymph node macrophages are ineffective in controlling S. pyogenes in vivo.

## Discussion

Thorough understanding of the route taken by extracellular bacteria from an extravascular site of infection into the bloodstream is currently lacking. Severe bloodstream infection is commonly believed to arise after invasion of blood vessel endothelium by bacteria at the site of local infection. However, the incidence of devastating metastatic infections that lack a clear portal of entry, such as those caused by S. pyogenes[41,42], is indicative that other routes of entry may be important, and underlines the need for more comprehensive investigation of the issue. In this article, we have revealed that extracellular lymphatic metastasis is a major mechanism of systemic dissemination in S. pyogenes infection, demonstrating that the lymphatics serve as both a survival niche and a preferential conduit to reach the bloodstream for S. pyogenes.

In humans, common manifestations of S. pyogenes infections, such as streptococcal tonsillitis and pharyngitis, are well-recognised to lead to lymphadenitis[11,43] and S. pyogenes DNA can be found in the lymph nodes of such patients[44]. We propose that metastatic bacterial spread to and, crucially, *through* draining lymph nodes during infections of the upper respiratory tract and tonsils, or other sites, provides an explanation for the phenomenon of occult bacteraemia and distant infection related to seemingly innocuous or non-invasive infections that lack an apparent portal of entry[41,42]. Notably, lymphatic metastasis was enhanced in hypervirulent isolates of S. pyogenes that are associated with severe clinical outcomes, implying its importance in human streptococcal disease. In these hypervirulent strains, we found that elevated expression of hyaluronan capsule (historically described as hyaluronic acid or hyaluronate capsule) and SpyCEP, augmented by mutations in the CovR/S regulatory system, increased bacterial burden within the lymphatic system in experimental disease. However, invasive S. pyogenes infections are rare and patently the lymphatic metastasis of bacteria does not routinely allow every trivial infection to develop into a serious systemic disease. The exact circumstances under which lymphatic metastasis can drive fatal S. pyogenes infection in humans remain undefined, but would be expected to involve a combination of host and environmental factors, including

underlying medical conditions[45], prior injury[34,46], level of exposure to *S. pyogenes*[47] and host genetic factors[48–50], as well as strain-specific virulence.

Beyond invasive disease, our data showed that *S. pyogenes* strains of lesser virulence could also reach and persist within lymph nodes, but without driving development of intense bacteraemia. Presence of bacteria in lymph nodes and the resulting inflammation has been shown to disrupt immune responses[51–53]. *S. pyogenes*, additionally, can secrete immune-dysregulating proteins termed superantigens that activate T cells in an uncontrolled manner[54] to disrupt immune responses in lymphoid tissue[55,56]. Bacterial lymphatic metastasis may contribute to recurrent childhood streptococcal infections[57] by slowing natural development of protective immunity[57]. Moreover, heavily encapsulated *S. pyogenes* isolates are associated with the autoimmune disease rheumatic fever[58]; strong lymphatic-retention of these strains mediated in part by hyaluronan capsule-mediated sequestration by LYVE-1, as demonstrated here, could provide a persistent stimulus that may lead to development of autoimmunity, driven in part by superantigens. Additionally, the observed interaction between streptococci and lymphocytes within the efferent lymphatics, possibly mediated through bacterial adhesion to CD44[59], could be anticipated to have consequences for immunity. Nevertheless, there is also potential for therapeutic exploitation of lymphatic bacterial homing, through heterologous expression of hyaluronan capsule in bacterial vaccine vectors to enhance antigen delivery to lymph nodes.

A small number of recent studies have shown that injected bacteria that do reach the local draining lymph node are constrained by immune responses and either reported that the bacteria did not reach sequential draining lymph nodes or did not investigate them[5,35,53]. Although some of our observations may be specific to the strain of mouse or specific hypervirulent strains used, our experiments with *S. pyogenes* found lymphatic metastasis to occur in several common strains of mouse, and in several different clinically relevant isolates. Experimental ablation of innate immune cells in lymph nodes has been demonstrated to increase systemic burden for some bacteria, although these studies did not characterise the route of spread[5,35]. CD169[+] subcapsular and medullary sinus macrophages are strategically positioned in lymph nodes and can prevent systemic spread of certain viruses[38,39]. Though a similar role has been suggested in bacterial elimination, the data are less clear[5,35]. *S. pneumoniae* has recently been reported to replicate inside analogous spleen CD169[+] macrophages[60] and our results further temper the significance of CD169[+] macrophages in controlling dissemination of virulent bacteria. Contrasting with intracellular replication of *S. pneumoniae* observed in the spleen[60], we observed that encapsulated hypervirulent *S. pyogenes* were not sequestered by lymph node macrophages, but instead replicated extracellularly. Neutrophils are recruited to local draining lymph nodes close to an initial infection site and play a vital role in combating bacterial infection[5,35,61]. We found that, in addition to local draining lymph nodes, large numbers of neutrophils were also recruited to distant draining lymph nodes far from the initial infection site. *S. pyogenes* was able to undermine this response by resisting phagocytosis and impairing neutrophil-recruitment through production of the chemokine-cleaving protease, SpyCEP, a finding that supports a key role for the chemokine CXCL8 (CXCL1/2 in mice) in directing neutrophil migration[36] towards live bacteria within lymph nodes. Thus, in the case of virulent *S. pyogenes*, neither neutrophils nor macrophages can contain bacterial infection effectively within lymph nodes.

While *S. pyogenes* is unique in the extent of its association with bacterial lymphangitis and lymphadenitis in humans, other species of bacteria have been recovered from lymph nodes in patients undergoing surgery[62–64], with some evidence that lymph nodes may serve as a route of bacterial dissemination[65]. Hyaluronan, in addition to mediating specific binding to lymphatic vessels, is a large polyanionic and hygroscopic molecule that can serve to mask underlying adhesion molecules on the bacterial surface[66,67], inhibit phagocytic clearance[68] and increase electrostatic repulsion in the extracellular matrix[69]: features that enhance the uptake of particles into lymphatic vessels[70,71]. Many other bacterial species produce exopolysaccharides (including capsules), which can confer similar non-specific physical properties to those of hyaluronan. Additionally, as most bacteria are of a size that permits passive lymphatic transport, lymphatic metastasis could be pertinent to the pathogenesis of other bacterial species.

Earlier investigators provided evidence that bacteria were capable of passage through the lymphatics[72], but these studies did not address the resulting impact on the development of infection, and technical aspects of their work have been queried[73]. Furthermore, more direct routes of bloodstream invasion have been reported for extracellular bacteria in different models[73–75]. Recently, elucidation of a multitude of bacterial cellular-invasion mechanisms[75–77], often involving intracellular survival, has further shifted focus away from the role of the lymphatics in extracellular bacterial dissemination. As such, contemporary research on translocation of bacteria in the lymphatics has focused on the concept of Trojan horses: migrating phagocytes that carry internalised, but viable, bacteria[8,78]. The findings herein unequivocally establish that *S. pyogenes* remains extracellular while disseminating from a local site of infection through multiple sequential draining lymph nodes to reach the bloodstream via efferent postnodal lymphatic vessels. Notwithstanding any immediate relevance to streptococcal diseases, the propagation of bacteria within the lymphatic system has broad relevance to host-pathogen interactions and immunity. The results of this study, demonstrating the importance and mechanisms of bacterial lymphatic metastasis, prompt reconsideration of currently understood paradigms of bacterial dissemination and pathogenesis.

## Methods

**Bacterial strains and growth conditions**. *Streptococcus pyogenes* (Supplementary Table 1) was cultured on Columbia horse blood agar (CBA) (EO Labs) or in Todd Hewitt broth with yeast extract (both Oxoid) at 37 °C, in 5% $CO_2$. *Lactococcus lactis* were grown on CBA at 30 °C in air. For mutagenesis protocols, LB media (Oxoid) were used to grow *Escherichia coli* at 37 °C in air. Growth media were supplemented with 50 µg/ml kanamycin for selection and maintenance of pUCMUT-containing *E. coli* and 400 µg/ml kanamycin for selection of gene-disruption *S. pyogenes* mutants. Previously derived SpyCEP expression vector pCepA[79] or empty vector control pDestErm were maintained in *L. lactis* with 5 µg/ml erythromycin.

**Construction of S. pyogenes gene-disruption mutants**. A *S. pyogenes* hyaluronan capsule knockout mutant (H1454) of isolate H584 was constructed by disruption of *hasA* and *hasB* using the suicide vector pUCMUT[80,81]. A 500 bp fragment of the 5′ *hasA* gene was amplified (forward primer: 5′- GGGGTACCTATCTTGATTTAT CTAAATATG-3′, reverse primer: 5′- GGAATTCGTTTCTAGCATTCAAATGTC CT-3′) incorporating EcoRI and KpnI restriction sites into the 5′ and 3′ ends respectively, and a 500 bp fragment of the 3′ *hasB* gene was amplified (forward primer: 5′-ACGCGTCGACATGATGATCGAATAGGAATGC-3′, reverse primer: 5′- AACTGCAGCAATCATACCACCAACTGCAG-3′) incorporating PstI and SalI restriction sites into the 5′ and 3′ ends respectively, and cloned into digested pUCMUT either side of the *aph3* kanamycin resistance gene. The construct was introduced into *S. pyogenes* by electroporation and crossed into the chromosome by homologous recombination. PCR analysis demonstrated a double recombination event between chromosomal *hasA* and *hasB* and the corresponding 500 bp fragments which abrogated production of hyaluronan capsule. To construct a SpyCEP allelic replacement mutant (H1567), in hypervirulent strain H1565, a 520 bp fragment of the *cepA* gene and a 485 bp fragment of the intergenic region between *cepA* and hypothetical protein M5005_Spy0343 were amplified and cloned into the suicide vector pUCMUT[81] to produce pUCMUT*cepA* using appropriate sequential restriction enzymes (forward primer: 5′-ATAGCGAATTCAACAGA-CAAAAACCTCCGCT -3′, reverse primer: 5′-ATAGGGTACCTTGTGATTGCT

CTTTTTCACGA-3′, incorporating EcoRI and KpnI restriction sites into 5′ and 3′ ends, respectively, and forward primer: 5′-AGACGTCGACCAAAGCGCAAAGCGACAACAAA-3′, reverse primer: 5′-ATAGCTGCAGAACTGGGGGTGTCTCCGT-3′, incorporating SalI and PstI restriction sites into the 5′ and 3′ ends). Plasmids were propagated in *E. coli* XL-10 gold (Agilent) and the construct electroporated into *S. pyogenes* H1565 and inserted into the chromosome by homologous recombination. PCR analysis demonstrated a double recombination event between chromosomal *cepA* and the 520 bp and 485 bp fragment. The allelic replacement fully negated CXCL8 cleavage activity and production of SpyCEP, measured by CXCL8 cleavage assays and western blots, respectively.

**Mouse-passaged derivatives of *S. pyogenes* H584**. Derivatives of H584 with a mutation in the CovR binding site of the hyaluronan capsule promoter P2 (H1458) that resulted in elevated hyaluronan, or a *covR* mutation (H1565) were recovered from a lymph node and spleen respectively after one in vivo passage and have been previously defined[34].

**Mice**. In vivo experiments were performed in accordance with the Animal (Scientific Procedures) Act 1986, with appropriate UK Home Office licenses according to established institutional guidelines using 4–10-week-old FVB/n, or where stated BALB/c and C57BL/6 female mice (Charles River UK). Mice were housed in standard conditions, with an ambient temperature of 21 °C, 50% humidity and a 12-h light cycle.

**Analysis of lymphatic drainage**. Mice were injected intramuscularly into the hindlimb with 50 μl of 1% Evans blue dye in PBS, which highlights draining lymphatic vessels and lymph nodes. After 30 min, mice were euthanised, dissected, and photographed.

**Infections**. Mice were challenged intramuscularly with $10^4$–$10^8$ CFU of *S. pyogenes* in 50 μl PBS and quantitative endpoints compared at 0.5, 3, or 24 h post-infection. To maintain SpyCEP expression vector pcepA and empty vector control pDestErm in *L. lactis* in vivo, both groups of *L. lactis*-infected mice were given 100 μl of 100 μg/ml erythromycin intraperitoneally 4 h before infection.

**Enumeration of bacteria from organs and blood**. At specified endpoints, mice were anaesthetised with isoflurane (4% induction, 2% maintenance), blood taken by cardiac puncture into tubes containing heparin, then euthanised. Thigh muscle, liver, spleen, and ipsilateral and contralateral axillary, brachial, iliac, and inguinal lymph nodes were dissected for further analyses. Dissected lymph nodes were first disrupted with a motorised pellet pestle, and all organs and tissues were homogenised with scissors in PBS. Bacterial CFU counts were determined by plating of homogenised tissue and blood samples onto the specified agar, with or without dilution in PBS, as appropriate. CFU are displayed as per ml of blood, per g of liver, per leg, or per organ for all figures.

**Preparation of single-cell suspensions from organs**. Dissected lymph nodes were disrupted with a motorised pellet pestle and then chopped further with scissors. Dissected spleens were minced with scissors. Lymph node and spleen fragments were then digested for 20 minutes at 37 °C with 200 U/ml collagenase IV and 100 U/ml DNase I (both Worthington Biochemical Corp) in digestion buffer (HBSS with 0.5% BSA, 2 mM $Ca^{2+}$ and 10 mM HEPES). Spleens were next passed through a 100 μm cell strainer in cell buffer (HBSS with 2 mM EDTA, 0.5% BSA, and 10 mM HEPES) containing 100 U/ml DNase I. For lymph nodes, large tissue fragments were allowed to settle and the supernatant, which contained the cells of interest, was removed and the cells washed before resuspension in cell buffer. If required, erythrocytes in spleen suspensions and blood were lysed with BD Pharm Lyse (BD Biosciences). If bacteria were to be maintained in the sample, a high-speed spin ($10,000 \times g$) of supernatants was performed after mouse cells were removed with a lower speed spin ($400 \times g$) and the two fractions subsequently recombined.

**Ex vivo gentamicin protection assays**. Single-cell suspensions from the organs of mice infected with *S. pyogenes* were incubated with or without 100 μg/ml of gentamicin in HBSS (with 10 mM HEPES) for 1 h at 37 °C in 5% $CO_2$. Cells and any bacteria were then washed to remove residual antibiotic, using the two-step centrifugation approach described above, and the mouse cell and bacterial fractions recombined. Samples were then resuspended in pH 11 $H_2O$, to efficiently lyse murine cells[82], and plated on CBA with dilution where required. CFU counts from gentamicin-treated samples were compared with untreated samples to calculate the percentage of intracellular bacteria. Alternatively, the assay was performed with immortalised bone-marrow-derived macrophages following a 3-h incubation with *S. pyogenes*.

**Flow cytometry on in vivo cell infiltrate**. Single-cell suspensions from the organs and blood of mice were resuspended in PBS and stained with Zombie NIR Viability dye (BioLegend). Cells were then washed and resuspended in staining buffer (HBSS with 2 mM EDTA, 0.5% BSA, and 5% rat serum) with a saturating concentration of anti-CD16/32 (93) and stained with anti-CD45-PerCP-Cy5.5 (30-F11), anti-CD11b-APC (M1/70), and anti-Ly6-G-PE (1A8) (all BioLegend). In other experiments, anti-CD11c-FITC (N418) (BioLegend), anti-CD169-PE (REA197) and anti-F4/80-FITC (REA126) (both Miltenyi Biotec) were used. Samples were analysed on an Attune NxT flow cytometer using Attune NxT software (Thermo Fisher). Cells were gated to determine live singlets, and neutrophils were defined as $CD45^+$, $CD11b^+$ and $Ly6G^+$.

**LYVE-1 functional blockade**. For LYVE-1 function blocking experiments, mice were injected intraperitoneally with 500 μg anti-LYVE-1 (mAb2125) or control rat IgG (both R&D) 24 h prior to intramuscular infection with *S. pyogenes*.

**Macrophage depletion**. To deplete macrophages in lymph nodes draining the infection site, mice were injected subcutaneously with 200 μl clodronate liposomes (7 mg/ml clodronate disodium), or the same volume of control PBS-liposomes (both FormuMax), or PBS. Injections were distributed between the tail (40 μl) and leg regions (160 μl), which drain to the inguinal lymph node, and performed 96 h prior to intramuscular infection with *S. pyogenes*.

**Macrophage culture**. Immortalised bone marrow-derived macrophages (iBMDM) were cultured in complete media (DMEM, with 10% FBS and 10 mM HEPES) at 37 °C in 5% $CO_2$. Prior to bacterial assays, macrophages were seeded near confluency onto tissue-culture treated multi-well plates and incubated overnight, before complete media was changed for warm HBSS (with $Mg^{2+}$ and $Ca^{2+}$ and 10 mM HEPES) to prevent bacterial growth during assays.

**Macrophage gentamicin protection assay**. *S. pyogenes* in HBSS was added to iBMDM adhered to tissue-culture treated multi-well plates at an MOI of 10:1. Plates were spun for 3 min at $500 \times g$ to synchronise infection and then incubated at 37 °C in 5% $CO_2$ for 3 h. The macrophage layer was then gently washed four times with warm PBS (with $Mg^{2+}$ and $Ca^{2+}$), before the addition of 100 μg/ml of gentamicin in HBSS for 1 h at 37 °C in 5% $CO_2$. Wells were again gently washed four times with warm PBS (with $Mg^{2+}$ and $Ca^{2+}$) to dilute the antibiotic and the macrophages were detached with Accutase (BioLegend) and lysed with 0.025% Triton X-100. Lysates were serially diluted in PBS and plated on CBA to measure intracellular CFU.

**Macrophage phagocytosis and viability assay**. *S. pyogenes* was stained with 10 μM Oregon Green 488-X, Succinimidyl Ester (Invitrogen) and added to iBMDM, prepared as above at an MOI of 10:1. Plates were spun for 3 min at $500 \times g$ to synchronise infection and then incubated at 37 °C in 5% $CO_2$ for 3 h. The macrophage layer was gently washed five times with warm PBS (with $Mg^{2+}$ and $Ca^{2+}$) to remove non-cell-associated *S. pyogenes* and then cells were detached with Accutase (BioLegend). Cells were resuspended in cold PBS with 2 mM EDTA and transferred to low-bind 96-well plates for staining with Zombie NIR Fixable Viability Dye (BioLegend). Macrophages were analysed using a Cytoflex flow cytometer using CytExpert (Beckman Coulter) and FlowJo (BD) for cell-associated and internalised *S. pyogenes*. Unstained *S. pyogenes* controls were used for gating, and a final concentration of 0.2% Trypan Blue or 25 μg/ml Ethidium Bromide, added two minutes before acquisition, was used to determine intracellular bacteria by quenching and/or Förster resonance energy transfer (FRET) of fluorescence from extracellular bacteria.

**Neutrophil phagocytosis assay**. Fresh human neutrophils were acquired from an approved subcollection of normal healthy donor blood samples of the Imperial College London Tissue Bank; all volunteers gave informed consent. Neutrophils were isolated from heparinised human blood using a MACSXpress Whole Blood Neutrophil Isolation Kit (Miltenyi Biotec) and resuspended in HBSS to $2 \times 10^6$ cells/ml. *S. pyogenes* was stained with 10 μM Oregon Green as above and added to neutrophils at an MOI of 10:1. Tubes were incubated at 37 °C with end-over-end mixing for 30 min. The reaction was stopped with an excess of cold PBS (with 5 mM EDTA) and cells were analysed on a FACSCalibur using BD CellQuest Pro (BD Biosciences) and FlowJo (BD). Controls using unstained *S. pyogenes* were used to set gating and a final concentration of 0.2% Trypan Blue added immediately prior to acquisition was determine intracellular bacteria by quenching and/or FRET of fluorescence from extracellular bacteria.

**Intravital imaging**. *S. pyogenes* were resuspended to an $OD_{600}$ of 10 (~$5 \times 10^9$ CFU/ml) in PBS, stained with far-red fluorescent DNA dye, SYTO 62 (final concentration 5 μM), and washed in PBS. In all, 4–6-week-old mice were infected intramuscularly with $10^8$ CFU of either unlabelled or fluorescently labelled bacteria and, following a holding period of 30 min, underwent deep terminal anaesthesia using a cocktail of ketamine (Ketamidor) and medetomidine (Domitor) injected intraperitoneally or subcutaneously. Mice were maintained at 37 °C on an electric heat-pad, and hair on the leg and flank was removed with an electric shaver and depilation cream. In all, 40 μl FITC-conjugated 2000 kDa dextran (2.5 mg/ml) was

injected subcutaneously into the tail and, in pilot experiments, 100 μl TRITC-conjugated 155 kDA dextran (2.5 mg/ml) (both Sigma) was injected intravenously. Intramuscular injection of SYTO 62 dye alone into the hindlimb did not stain host cells in the inguinal postnodal lymphatic, and injection of SYTO 62-stained *S. pyogenes* that did not reach the axillary node (H1454) showed no signal in the postnodal lymphatic (Supplementary Fig. 3d, e). Direct staining of host cells with SYTO 62 resulted in bright staining of the nucleus only, similar to DAPI (Supplementary Fig. 3f, g). For labelling of cells in efferent lymph, desired combinations of 5 μg of anti-B220 Brilliant Violet 421 (RA3-6B2), anti-CD3-PE (145-2C11), anti-CD11b-Alexa Fluor 488 (M1/70), anti-F4/80- Brilliant Violet 421 (BM8) (all BioLegend) and 100 μg Hoechst 33342 (Thermo Scientific) were injected subcutaneously into the tail. For imaging, anaesthetised mice were placed in a supine position and a vertical incision was made in the skin along the midline, extended laterally at the inguinal and axillary regions to create a skin flap. Mice were then moved into a prone position with the skin flap flat against a stage plate insert with central coverslip window. A piece of wood doweling was placed against the body of the mouse to help hold the skin flat and reduce the impact of breathing artifact during imaging (Supplementary Fig. 3b). Mice were transferred to the microscope stage and maintained at 37 °C in an incubation chamber (Life Imaging Services) throughout imaging. Isoflurane administered when required and mice were euthanised once imaging was finished. Light was generated from 405-nm, 488-nm, 561-nm, 594-nm and 633-nm lasers, as appropriate. Emitted light signal was detected using either a 10×/0.3, 20×/0.5, 25×/0.95, or 40×/1.1 objective lens, as required, using a Leica SP5 confocal microscope. Images were processed using LAS X (Leica) and ImageJ2.

**Immunofluorescence microscopy.** Lymph nodes were either fresh snap-frozen or fixed in 2% paraformaldehyde in PBS, followed by protection in 30% sucrose in PBS before snap-freezing. 6–15 μm frozen sections were cut using a cryostat, dried, fixed in cold acetone (-20 °C) for 5 min, and then blocked (5% FBS, 5% goat serum, 5% rat serum, and 0.5% BSA in PBS). When required, an avidin/biotin blocking system was used (BioLegend). Sections were then stained, as required, with combinations of anti-Ly6G-AF594 (1A8), anti-CD169-AF647 (3D6.112), biotinylated anti-PNAd-biotin (MECA-79) (all BioLegend), hamster anti-podoplanin (8.1.1) (eBioscience), anti-LYVE-1-eFluor 570 (ALY7) (Invitrogen), and anti-Group A Carbohydrate-FITC (9191) (Abcam) in staining buffer (blocking buffer with 0.05% Tween-20) for 1 h. Next, if required, slides were incubated with anti-hamster IgG-AF647 and Streptavidin-AF546 (both Invitrogen) in staining buffer for 45 minutes. Nuclei were counterstained with DAPI and sections mounted with ProLong Gold (both Invitrogen). Light was generated from 405-nm, 488-nm, 561-nm, 594-nm, and 633-nm lasers as required, and emitted light signal detected using either a 10×/0.3 or 20×/0.7 objective on a Leica SP5 confocal microscope. Tiled scans were performed to image entire lymph node sections and areas of special interest were imaged with a 40×/1.3 or 63×/1.4 objective. Mosaic images were generated from tiled stacks using LAS X (Leica) and zoom and panning videos of mosaic images generated using the IC FILM Zoom and Pan macro in ImageJ2. Orthogonal views and projections of confocal z-stacks were also constructed in ImageJ2.

**Statistics.** All statistical analyses were performed with GraphPad Prism 8.4. Comparison of two datasets was carried out using a two-tailed Student's *t*-test on $log_{10}$-transformed data or a two-tailed Mann–Whitney U test. For comparison of three or more data sets, one-way ANOVA followed by Tukey's multiple comparisons post-tests were performed on $log_{10}$-transformed data. A *p*-value of ≤ 0.05 was considered significant.

**Reporting summary.** Further information on research design is available in the Nature Research Reporting Summary linked to this article.

## Data availability

Source data are provided with this paper. Further data that support the findings of this study are available from the corresponding authors upon reasonable request.

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

## Acknowledgements

We thank David W. Holden for critical reading of the manuscript; James E. Moore and Simon Milling for helpful discussions; Andrew M. Tomlins for assistance with designing some graphics; Alanna Ruddell for advice on tail subcutaneous injections and drainage routes; Simon M. Rothery for creating the ImageJ2 Zoom and Pan macro for supplementary data presentation; Victor Nizet and Annelies S. Zinkernagel for *L. lactis* strains containing pDestErm and pCepA plasmids; and the laboratory of Paras K. Anand for supplying bone marrow-derived macrophages. Research reported was supported by Medical Research Council grant (MR/L008610/1) to S.S. and D.G.J. We also acknowledge the support of the NIHR Biomedical Research Centre at Imperial College and the NIHR BRC Tissue Bank. The contribution of N.N.L was supported by a Sir Henry Wellcome postdoctoral fellowship (103197/Z/13/Z) from the Wellcome Trust.

## Author contributions

Conceptualisation, M.K.S., D.G.J., N.N.L., and S.S.; methodology, M.K.S., L.E.L., C.E.T., L.A.J., N.N.L., K.W., and S.S.; investigation, M.K.S., L.E.L., M.P., L.A.J., S.B., K.K.H., and K.W.; writing—original draft, M.K.S., and S.S.; review and editing paper, M.K.S., K.W., D.G.J., and S.S.; funding acquisition, S.S. and D.G.J.

## Competing interests

The authors declare no competing interests.
