## [Peer Review File · Nature Communications]

Reviewers' comments:

Reviewer #1 (Remarks to the Author):

The authors present data showing that *S. pyogenes* remain extracellular while disseminating from a local site of infection through multiple sequential draining lymph nodes in order to reach the bloodstream via efferent postnodal lymphatic vessels. These data provide new insight and a new mechanism of systemic bacterial dissemination that does not rely on phagocytosis and intracellular transport by macrophages or neutrophils. The experiments are well performed and the data support the main conclusions of the manuscript. I have a few suggestions to help clarify and further strengthen the manuscript.

Comments:

Figure 3: Cortex. I would clarify that podoplanin is staining fibroblastic reticular cells in the cortex of the lymph node. Not all podoplanin positive cells are lymphatic endothelial cells.

Extended Fig 4. It looks like the IgG control group contains a potential outlier with 0 CFU burden that influenced strongly the statistics. Are these results really robust? Further, could there be a problem with delivery of LYVE-1 Ab to the targets? Could these experiments be performed in LYVE-1 knockout mice to be more definitive?

Page 11: Line 18: Clarify that beginning at this point in the paragraph you are referring back to the "wild-type" hypervirulent strain (H1565), not the deltaSpyCEP (H1567) strain. The paragraph is confusing on this point.

Fig 5a, it does not seem that SpyCEP plays a role in the systemic dissemination of the bacteria How do these data support or refute your mechanistic hypothesis?

Do you have data similar to the those in Fig 5e-h and Extended Fig 5 for the for the deltaSpyCEP (H1567) strain to contrast and show the role of SpyCEP in these mechanistic measurements? These data would strengthen the role of SpyCEP in the virulence and metastatic spread through lymphatic vessels. One would expect more phagocytosis of deltaSpyCEP bacteria compared to the wild type.

Minor Comments:

Page 22, Line 18: Check sentence.

Video 6, 7: “podoplanin (magenta) -positive subcapsular and medullary sinuses”: I would clarify that podoplanin is also staining fibroblastic reticular cells in the interior of the node. Not all podoplanin positive cells are lymphatic endothelium.

Reviewer #2 (Remarks to the Author):

An excellent work describing the initial phases of within host spread of *S. pyogenes*, with detailed investigation of early time points in multiple organs; importantly testing more than one bacterial strain. The surgical skill and the quality of microscopy, in particular intra-vital imaging is exceptional. The definition of the central role of the GAS hyaluronic acid capsule and in particular the bacterial chemokine protease in this phenotype is a great plus to this research work.

Specific comments:

Page 6 line 19 if you have no experimental data relative to CD44 mediated adhesion, please remove reference to Cd44 here and maybe refer to it with a reference in the discussion.

Figure 3 please describe in the legend what is stained by podoplanin and/or the relative part of the text on page 8 line 3-4.

Page 11 line 20 the observation of about a quarter of lymph node apparently occupied by GAS possibly lysing surrounding cells (Fig 5e), would probably warrant a stronger term than “prominent foci”. Is this an abscess like situation? Quite acute for an abscess, but neutrophil influx and cell lysis would point to an abscess or extended necrotising event.

Page 13 line 13 In the legend of video 9 it is stated that some bacteria appear to be localised within CD169+ cells and the video really appears to show this. It might be appropriate at least to mention it in the main text, as not everybody will read the legend to video 9.

Page 13 line 14 the lack of any effect of clodronate on the infection shows clearly lack of macrophage involvement. I may have missed it, but do the authors show that clodronate actually depleted the macrophages from the lymph nodes. Maybe a macrophage staining of a lymph node or spleen after clodronate treatment would clarify this.

Page 13 line 20. The conclusion of the sentence indicating that hypervirulent capsule/protease related phenotypes are responsible for macrophage inefficiency is not clear. In vivo normal GAS is

already evading macrophage control of lymph and you do not show any localised hyperproduction of capsule or protease by normal GASs. I therefore do not see the causative link to macrophage inefficacy in normal GAS infection.

Page 15 line 13 at the beginning of the paper you build all your story of lymphatic spread of GAS on specific interaction of the hyaluronic acid structure with LYVE-1. Now you are stating that any capsule, irrespective of its molecular structure, does the same job. This needs some better or additional explanation, as the relative paragraph in the discussion does not really clarify.

Reviewer #3 (Remarks to the Author):

In their manuscript, "Lymphatic metastasis of virulent extracellular bacteria drives systemic infection" (NComm19-38830), Siggins and colleagues report that *S. pyogenes* (SP) disseminate from a local site of infection via lymphatic conduits to a sequential series of draining lymph nodes (LN) and ultimately to the bloodstream. This transiting occurs via extracellular mechanisms (i.e. not shielded within phagocytes) and resists clearance by phagocytes. They identify streptococcal virulence mechanisms that facilitate this spread, particularly the hyaluronan capsule and a chemokine-cleaving protease, SpyCEP, which can inhibit PMN recruitment to a site of infection. They conclude by claiming this lymphatic spread mechanism is not unique to SP, but also is true for other "pathogenic extracellular bacteria" thereby invoking a "new perspective" in bacterial pathogenesis.

Major comments:

These investigators present convincing data that a local SP infection (mouse hindlimb) travels to an ipsilateral draining LN through a sequential series of LN to spleen and liver. Once the bacteria appear in the bloodstream at 24 hrs, the bacteria are detected in the contralateral LN. These data are supportive of their hypothesis that SP can disseminate beyond a local site of infection via lymphatics into the blood stream. These data were acquired by technically demanding lymphadenectomy of inguinal, iliac, axillary, and brachial LN on the ipsilateral and contralateral sides of 3-6 mice per group. They also used fluorescent SP and confocal intravital microscopy to demonstrate the SP were not within phagocytes as they traversed the lymphatics, and provided video support. They show that most of the leukocytes within the lymphatics were not CD11b phagocytes but rather CD3+ T cells and rarely entered the LN parenchyma. SP also were able to evade killing by CD169+ macrophages within LN.

My major concern is when these investigators expand their findings with SP to other "extracellular" bacteria and claim that this phenomenon is widespread and not a "quirk" of streptococcal pathogenesis. Here their data are far less robust. First, before they can make that statement, I would want to see many additional non-SP strains showing this behavior rather than one strain of each species. Their studies with SP appear to have a particular relevance in that with SP soft tissue

infection there is clinically evident lymphangitic spread, suggestive of lymphatic involvement. This is not the usual case with the Gram-negative bacteria (GNB) and perhaps for *S. aureus* as well. With regard to *S. aureus*, they are unable to show any spread at 6 hr (Fig 7) beyond the local site; however, USA 300 is very poorly virulent in mice and therefore would not be a good candidate for study. Thus, they cannot make any generalization about the role of capsule based on the confinement of USA300 (which has no capsule) to the injection site. With regard to the GNB (*Klebsiella*, *Pseudomonas* and *E. coli*) they state that these strains were clinical isolates obtained from the bloodstream of patients with soft tissue infections (see Methods—Bacterial Strains), but there is little additional information, including clinical description of the soft tissue infection. The strain designation in the absence of additional information on the capsule or O serotype provides no help. (Extended Data Table 1 lists two strains from “BioAID collection”—what is that??) They also do not offer any evidence of the role of virulence factors for these strains as they did for SP. For example, do we know anything about the capsules of either EC or KP (*Pseudomonas* does not have a capsular polysaccharide)? Does the anti-lymphatic endothelial receptor – (LYVE-1) also play a role for these other bacteria?

Having shown their technical expertise in delineating the lymphatics and isolating cellular elements within the lymphatics, it would have been quite illuminating for them to examine the lymphatic fluid. For example, is there immunoglobulin within the lymphatics?? In their discussion they state (p. 16, l.25) that the lymphatics do not “routinely allow every trivial bacterial infection” to develop into a serious systemic disease. Might they speculate on what types of protective mechanisms may come to bear? If SP secrete superantigens, why is there not a greater immune response within the LN given the large number of T cells and antigen-presenting cells?? Certainly if the authors suggest that the persistent stimulus of superantigen-secreting SP might induce autoimmune disease, there should be some evidence of immune response (e.g. germinal center formation).

Specific Comments:

Given my comments above, the title should be changed to substitute “*S. pyogenes*” in place of “extracellular bacteria”. They should also limit the conclusion in the abstract to SP and not “pathogenic extracellular bacteria”, and new perspectives on SP pathogenesis rather than “bacterial” pathogenesis.

Figure 2, panel “I”: to which cell surface are they referring?

p.8, Fig 3: Why do the SP not enter the LN parenchyma? It is difficult to see to what are the arrows pointing, and no indication is given in the figure legend. They should explain why they want to identify podoplanin—are they citing it as a specific lymphatic vessel marker or as a link between cardiovascular and lymphatic systems?

p. 9. Interestingly, the hyaluronan capsule’s role in virulence appears to be not so much in evading phagocytes but in retaining SP within the lymphatics based on their interaction with LYVE-1—is that correct? If so, this would be different from the role of capsule in *Klebsiella* and *E. coli*. Is there any evidence of germinal center formation within the lymph nodes transited by the SP? How did the SP leave the lymphatics at 24 hr and enter the LN parenchyma??

p.11. Was there any effect of SpyCEP mutants on PMN recruitment to the original hindlimb site of infection??

p. 20., Methods. It appears that the bacteria were grown overnight before use and hence in stationary phase. Was there any comparison of bacteria grown to log versus stationary phase??

Mice: In their methods they claim to use FVB/n, BALB/c and C57BL/6 mice, yet nowhere in their figure legends do they indicate the strain of mice. And why the three different strains??

Author Response to Peer Review: (NComm19-38830)

Siggins *et al*: Lymphatic Metastasis of Virulent Extracellular Bacteria Drives Systemic Infection

Reviewer #1 (Remarks to the Author):

The authors present data showing that S. pyogenes remain extracellular while disseminating from a local site of infection through multiple sequential draining lymph nodes in order to reach the bloodstream via efferent postnodal lymphatic vessels. These data provide new insight and a new mechanism of systemic bacterial dissemination that does not rely on phagocytosis and intracellular transport by macrophages or neutrophils. The experiments are well performed and the data support the main conclusions of the manuscript. I have a few suggestions to help clarify and further strengthen the manuscript.

Author response: Thank you for this positive summary of our work and for your helpful comments. We have made the changes suggested and agree that they further strengthen the manuscript. Point by point responses and details are given below.

Comments:

Figure 3: Cortex. I would clarify that podoplanin is staining fibroblastic reticular cells in the cortex of the lymph node. Not all podoplanin positive cells are lymphatic endothelial cells.

Author Response: Thank you for this suggestion, we fully agree and should have pointed this out earlier. The matter is now clarified with the sentence: 'Podoplanin staining highlights lymphatic endothelial cells in the subcapsular and medullary sinuses, as well as fibroblastic reticular cells in the cortex of the lymph node', which appears in the legend of Fig. 3.

Extended Fig 4. It looks like the IgG control group contains a potential outlier with 0 CFU burden that influenced strongly the statistics. Are these results really robust?

Author Response: We apologise that the original formatting of this graph obscured several data points and gave a misleading impression of the dataset. This has been rectified and *Extended Data Fig. 4* now more clearly shows the reader that there were multiple mice with 0 CFU values. Such '0' values are commonly observed using this model and we have no reason to suspect they are outliers. We have performed similar experiments before and obtained comparable results such as shown in *Figure R1*. Furthermore, the choice of statistical test employed (*Mann-Whitney rank sum U test*) minimises the impact of these '0' values, as it only compares the mean of ranks, rather than the actual values. Hence, we are confident that the data are robust and reproducible.

Further, could there be a problem with delivery of LYVE-1 Ab to the targets? Could these experiments be performed in LYVE-1 knockout mice to be more definitive?

Author Response: In addition to previously performing a similar experiment with LYVE-1 blocking antibody (Fig. R1), we have also used LYVE-1 knockout mice for the same experiment (Fig. R2). Both methods gave similar results – albeit with a different bacterial strain – that are fully in line with the data obtained for this manuscript. Therefore, we are confident that the antibody delivery is sufficient and the results definitive. The original experiments shown in Fig. R1 and R2 were performed prior to our appreciation of the mechanism of *S. pyogenes* lymphatic metastasis, and so at the time we did not assess the distant draining axillary lymph node, which necessitated repetition with the inclusion of the axillary lymph node in this work. The findings of these experiments were reported in *Lynskey et al*¹

Fig. R1. LYVE-1 functional blockade reduces GAS dissemination to draining lymph nodes. Dissemination of M18 GAS in murine soft-tissue infection following LYVE-1 mAb blockade (n = 22/group). Quantitative culture of GAS at site of infection (A), ipsilateral draining LN (B), spleen (C) and blood (D) 3 h post-infection. Lines depict median values in each case (Mann Whitney; * = p < 0.05, ** = p < 0.01). (Control antibody group: circles indicate isotype control antibody and triangles indicate polyclonal control IgG).

Fig R2. Genetic deletion of LYVE-1 impairs GAS dissemination to draining lymph nodes. Dissemination of M18 GAS in murine soft-tissue infection in constitutive LYVE-1^{-/-} mice (n = 7/group). Numbers of GAS recovered at site of infection (A), ipsilateral draining LN (B), spleen (C) and blood (D) 3 h post-infection. Lines depict median values in each case (Mann Whitney; * = p<0.01).

Page 11: Line 18: Clarify that beginning at this point in the paragraph you are referring to the “wild-type” hypervirulent strain (H1565), not the deltaSpyCEP (H1567) strain. The paragraph is confusing on this point.

Author Response: Thank you for pointing out this potential ambiguity, this has now been clarified.

Fig 5a, it does not seem that SpyCEP plays a role in the systemic dissemination of the bacteria How do these data support or refute your mechanistic hypothesis?

Author Response: We should have made clear that the inability to show a clear effect on SpyCEP in systemic spread is likely due to the influence of other virulence factors expressed by hypervirulent *S. pyogenes* which serve to mask the impact of SpyCEP. For example, hyaluronan capsule and M1 protein both inhibit phagocytosis by neutrophils and are both over-expressed by hypervirulent ΔSpyCEP *S. pyogenes*. To verify this assertion, we used a previously employed expression plasmid to heterologously express SpyCEP in *Lactococcus lactis*: an avirulent bacterium that is readily phagocytosed, lacks virulence factors and does not naturally express SpyCEP. When *L. lactis* with an empty control plasmid was injected into mice, the bacteria were confined to the injection site and did not disseminate. In clear contrast, *L. lactis* expressing SpyCEP showed striking dissemination to draining lymph nodes and systemic organs (Extended Data Fig. 5i and discussed: Page 12; lines 17–21). Taken together, SpyCEP plays a clear role in combatting neutrophil recruitment to the lymph node and permitting bacterial expansion, and the relative contribution of SpyCEP to systemic dissemination will be strain-dependent.

Do you have data similar to the those in Fig 5e-h and Extended Fig 5 for the for the deltaSpyCEP (H1567) strain to contrast and show the role of SpyCEP in these mechanistic measurements? These data would strengthen the role of SpyCEP in the virulence and metastatic spread through lymphatic vessels. One would expect more phagocytosis of deltaSpyCEP bacteria compared to the wild type.

Author Response: Thank you for this suggestion. We now include images from mice infected with hypervirulent or SpyCEP-deficient *S. pyogenes* in Extended Data Fig. 5. The low magnification images (panels d and g) demonstrate the greater bacterial: neutrophil ratio in mice infected with the hypervirulent SpyCEP+ strain compared with those infected with the SpyCEP-deficient strain. In

contrast to mice infected with the hypervirulent parent strain, comparatively strong recruitment of neutrophils was observed in mice infected with SpyCEP-deficient *S. pyogenes*. Nevertheless, the bacteria were not completely controlled within the lymph node. As discussed above, the capability of SpyCEP to drive bacterial systematic spread was apparent when expressed in *L. lactis* (an avirulent, readily phagocytosed bacterium) (*Extended Data Fig. 5j*). Hence, SpyCEP does indeed affect both lymphatic dissemination and systemic bacterial burden, but this effect will vary depending on bacterial strain.

Minor Comments:

Page 22, Line 18: Check sentence.

Author Response: Thank you, this sentence has been corrected.

Video 6, 7: “podoplanin (magenta) -positive subcapsular and medullary sinuses”: I would clarify that podoplanin is also staining fibroblastic reticular cells in the interior of the node. Not all podoplanin positive cells are lymphatic endothelium.

Author Response: Thank you for this suggestion. This is now explained with the sentence: “Podoplanin staining in the interior of the node is due to the presence of fibroblastic reticular cells” in the legends for *Videos 6 and 7*.

Reviewer #2 (Remarks to the Author):

*An excellent work describing the initial phases of within host spread of *S. pyogenes*, with detailed investigation of early time points in multiple organs; importantly testing more than one bacterial strain. The surgical skill and the quality of microscopy, in particular intra-vital imaging is exceptional. The definition of the central role of the GAS hyaluronic acid capsule and in particular the bacterial chemokine protease in this phenotype is a great plus to this research work.*

Author Response: Thank you for this extremely positive summary of our work and for your encouraging compliments regarding its quality.

Specific comments:

Page 6 line 19 if you have no experimental data relative to CD44 mediated adhesion, please remove reference to Cd44 here and maybe refer to it with a reference in the discussion.

Author Response: Reference to CD44 has been removed from the introduction. Instead, CD44 is now referred to in a relevant section within the discussion (*Page 20; lines 1–2*)

Figure 3 please describe in the legend what is stained by podoplanin and/or the relative part of the text on page 8 line 3-4.

Author Response: Thank you for this suggestion. A description: *'Podoplanin staining highlights lymphatic endothelial cells in the subcapsular and medullary sinuses, as well as fibroblastic reticular cells in the cortex of the lymph node'* has been added to the legend for Fig. 3.

Page 11 line 20 the observation of about a quarter of lymph node apparently occupied by GAS possibly lysing surrounding cells (Fig 5e), would probably warrant a stronger term than "prominent foci". Is this an abscess like situation? Quite acute for an abscess, but neutrophil influx and cell lysis would point to an abscess or extended necrotising event.

Author Response: Thank you for this helpful suggestion. We fully agree that there appears to be extended necrosis and that our previous description of the infected lymph node understated the degree of streptococcal invasion. Accordingly, we have now changed the text to: *'extensive prominent foci of invading bacteria were evident deeper in the lymph node parenchyma, often forming large necrotic regions'*. (*Page 12; lines 2–4*).

Page 13 line 13 In the legend of video 9 it is stated that some bacteria appear to be localised within CD169+ cells and the video really appears to show this. It might be appropriate at least to mention it in the main text, as not everybody will read the legend to video 9.

Author Response: Thank you, we agree. The main text now includes reference to this: *'The overwhelming majority of S. pyogenes were neither sequestered nor internalised by CD169+ cells, with only extremely rare instances of colocalization'*. Our analysis of many sections from numerous lymph nodes indicates that the bacteria are overwhelmingly extracellular. However, the high magnification required to discern sufficient detail of internalization, means that it is not possible to give a truly representative image or video.

Page 13 line 14 the lack of any effect of clodronate on the infection shows clearly lack of macrophage involvement. I may have missed it, but do the authors show that clodronate actually depleted the macrophages from the lymph nodes. Maybe a macrophage staining of a lymph node or spleen after clodronate treatment would clarify this.

Author Response: Thank you for this suggestion, macrophage straining of a lymph node after clodronate treatment is now included in *Extended Data Fig. 6*.

Page 13 line 20. The conclusion of the sentence indicating that hypervirulent capsule/protease related phenotypes are responsible for macrophage inefficiency is not clear. In vivo normal GAS is already evading macrophage control of lymph and you do not show any localised hyperproduction of capsule or protease by normal GASs. I therefore do not see the causative link to macrophage inefficiency in normal GAS infection.

Author Response: We agree. The original wording did not accurately convey our intended generalised point about ineffective macrophage control of *S. pyogenes*. Therefore, it has been replaced with the following: *'In co-culture, M1T1 wildtype H598 and otherwise isogenic highly encapsulated ΔP2 and CovR^{mutant} strains showed only limited internalisation by murine bone marrow-derived macrophages (Extended Data Fig. 6c, d). Moreover, these bacteria all induced significant macrophage cell death (Extended Data Fig. 6e), supporting the observation that lymph node macrophages are ineffective in controlling S. pyogenes in vivo.*

Page 15 line 13 at the beginning of the paper you build all your story of lymphatic spread of GAS on specific interaction of the hyaluronic acid structure with LYVE-1. Now you are stating that any capsule, irrespective of its molecular structure, does the same job. This needs some better or additional explanation, as the relative paragraph in the discussion does not really clarify.

Author Response: Thank you for pointing out this ambiguity. We now have improved the clarity of our explanation in the text. Briefly, hyaluronan influences lymphatic spread of *S. pyogenes* in two distinct ways. The first of these is mediated by selective capture of the hyaluronan capsule by LYVE-1, which enhances retention of *S. pyogenes* in lymph nodes. This specific effect is exclusive to hyaluronan and thus *S. pyogenes*. The second, non-specific way by which hyaluronan may influence lymphatic dissemination is *through its biophysical properties, i.e. its polyanionic and hygroscopic – water attracting – nature, completely independent of its LYVE-1 binding capacity*. Based on studies with particles, such properties are predicted to enhance lymphatic uptake of *S. pyogenes* and impair its phagocytosis. Other capsule types with broadly similar physical properties to hyaluronan would also be expected to confer similar effects on bacteria by means of the same 'non-specific', LYVE-1 independent mechanisms.

In the main text we now state: *'Though streptococcal-specific virulence factors, **such as hyaluronan**, play a major role in enhancing lymphatic dissemination, other bacteria produce different virulence*

mediators that could have similar effects and are of a size that could enable passive lymphatic transit'. (Page 15 and discussed page 21; lines 3–22).

Reviewer #3 (Remarks to the Author):

*In their manuscript, “Lymphatic metastasis of virulent extracellular bacteria drives systemic infection” (NComm19-38830), Siggins and colleagues report that *S. pyogenes* (SP) disseminate from a local site of infection via lymphatic conduits to a sequential series of draining lymph nodes (LN) and ultimately to the bloodstream. This transiting occurs via extracellular mechanisms (i.e. not shielded within phagocytes) and resists clearance by phagocytes. They identify streptococcal virulence mechanisms that facilitate this spread, particularly the hyaluronan capsule and a chemokine-cleaving protease, SpyCEP, which can inhibit PMN recruitment to a site of infection. They conclude by claiming this lymphatic spread mechanism is not unique to SP, but also is true for other “pathogenic extracellular bacteria” thereby invoking a “new perspective” in bacterial pathogenesis. Major comments: These investigators present convincing data that a local SP infection (mouse hindlimb) travels to an ipsilateral draining LN through a sequential series of LN to spleen and liver. Once the bacteria appear in the bloodstream at 24 hrs, the bacteria are detected in the contralateral LN. These data are supportive of their hypothesis that SP can disseminate beyond a local site of infection via lymphatics into the blood stream. These data were acquired by technically demanding lymphadenectomy of inguinal, iliac, axillary, and brachial LN on the ipsilateral and contralateral sides of 3-6 mice per group. They also used fluorescent SP and confocal intravital microscopy to demonstrate the SP were not within phagocytes as they traversed the lymphatics, and provided video support. They show that most of the leukocytes within the lymphatics were not CD11b phagocytes but rather CD3+ T cells and rarely entered the LN parenchyma. SP also were able to evade killing by CD169+ macrophages within LN.*

Author Response: Thank you for your positive evaluation of our work. Our detailed point by point responses to your concerns follow below.

My major concern is when these investigators expand their findings with SP to other “extracellular” bacteria and claim that this phenomenon is widespread and not a “quirk” of streptococcal pathogenesis. Here their data are far less robust. First, before they can make that statement, I would want to see many additional non-SP strains showing this behavior rather than one strain of each species.

Author response: Our primary purpose was indeed to undertake a detailed investigation of lymphatic dissemination of *S. pyogenes*- precisely because of the clinical observation that bloodstream infections sometimes develop with no obvious portal of entry, and because of the manifest association between *S. pyogenes* and lymphadenitis and lymphangitis. Nonetheless, it would be disingenuous to ignore the possibility that other bacteria might utilise similar routes; the

size of bacteria renders passive transit in lymphatic fluid likely yet it is a niche that has been under-investigated. The bacteria used were mostly clinical strains that had caused bloodstream infections; we do not necessarily expect all strains of these other species to survive lymphatic spread.

Nevertheless, in response to the Reviewer's concerns over this issue, we have expanded our analysis of lymphatic spread of non-*S. pyogenes* bacteria (Fig. 7). The figure now includes data from 13 unique bacterial strains that represent 6 different species of non-*S. pyogenes* extracellular bacteria (3 Gram positive, 3 Gram negative). This is in addition to examination of 6 *S. pyogenes* clinical isolates and 5 isogenic mutant strains. Furthermore, to maximise relevance to human disease, we focussed on use of clinical, mostly bacteraemia isolates (CC15 and C22 strains of *S. aureus*; ST88 and ST131 strains of *E. coli*; and serotype Ia and V strains of *S. agalactiae* (GBS) (Fig. 7 and Extended Data Fig. 5i). The data show that lymphatic-dominated spread can occur early during infection with a diverse range of bacteria; whether such dissemination has clinical consequences is unclear but could be of importance when considering development of cognate immunity or autoimmunity. We hope the Reviewer will understand that such investigations are beyond the scope of the current work.

Their studies with SP appear to have a particular relevance in that with SP soft tissue infection there is clinically evident lymphangitic spread, suggestive of lymphatic involvement. This is not the usual case with the Gram-negative bacteria (GNB) and perhaps for S. aureus as well.

Author response: *S. pyogenes* was chosen as the focus of the study precisely because of this clinically evident association. We agree that other extracellular bacteria do not exhibit this feature in humans, but, as outlined above, we felt that it was important to determine if other bacteria might demonstrate the ability to enter the lymphatic system and survive the early stages of an experimental infection. We have now highlighted the distinction between *S. pyogenes* (that does have a relationship with clinical lymphatic syndromes) and other bacteria (that do not) in the manuscript; the distinction may well related to the unusual interaction of *S. pyogenes* with LYVE-1 or a host-specific difference in ability to clear different types of bacteria.

Notwithstanding these caveats, viable Gram negative bacteria have been recovered from human lymph nodes in patients undergoing surgery for cancer²⁻⁶ and were associated with increased risk of postoperative sepsis, though the routes of dissemination are poorly understood^{4,6}. This is discussed briefly: (Page 21; lines 18–21). As such, there is some clinical evidence that lymphatic spread of other bacteria could be of relevance to humans.

It should be noted that the use of a soft tissue intramuscular hindlimb infection model across all bacteria is to allow analysis of a defined route of lymphatic drainage, as well as to give comparative consistency to experiments. We are not suggesting that all bacteria assessed would naturally first

cause soft tissue infection and then routinely disseminate through the lymphatics to drive systemic spread. The focus of these experiments is to assess whether these bacteria, once given partial access to the draining afferent lymphatics (via i.m. injection), are able to pass through sequential draining lymph nodes in order to drive systemic infection. In natural infection, initial access to the lymphatics could involve many different routes. For example, during tonsillitis, *S. pyogenes* could transit in efferent lymph from the palatine tonsil to tonsillar lymph nodes and beyond.

With regard to S. aureus, they are unable to show any spread at 6 hr (Fig 7) beyond the local site; however, USA 300 is very poorly virulent in mice and therefore would not be a good candidate for study.

Author response: We thank the Reviewer for making this valid point and have now undertaken assessment of lymphatic dissemination using two additional *S. aureus* strains (Fig. 7f), both of which are clinical bloodstream isolates from STs that represent the top five causing bacteraemia in the UK. These strains were able to reach local draining lymph nodes more efficiently than USA300, though still exhibited poor metastatic ability overall, compared to most other bacteria assessed. The text has been amended to describe these important new data and the data included in a fully revised Fig. 7 (Discussed Page 16; lines 3–7).

Thus, they cannot make any generalization about the role of capsule based on the confinement of USA300 (which has no capsule) to the injection site.

Author response: Thank you for highlighting this. On reflection, we agree that our comments regarding the contribution of capsule in the poor lymphatic spread of USA300 were overstated in the original manuscript. Accordingly, to address the Reviewer's concern, reference to the capsule of USA300 has been removed from the manuscript as a rationale for the findings.

With regard to the GNB (Klebsiella, Pseudomonas and E. coli) they state that these strains were clinical isolates obtained from the bloodstream of patients with soft tissue infections (see Methods—Bacterial Strains), but there is little additional information, including clinical description of the soft tissue infection. The strain designation in the absence of additional information on the capsule or O serotype provides no help. (Extended Data Table 1 lists two strains from “BioAID collection”—what is that??)

Author response: We apologise for not providing adequate information about the strains used. The Bioresource in Adult Infectious Diseases (BioAID) is a multicentre UK NIHR Bioresource, collecting biological samples from patients who present to the Emergency Department with suspected infection⁷. The cohort study protocol is now referenced in the manuscript. *Extended Data Table 1* has been updated to include the additional contemporary clinical strains used in response to the

Reviewer's comments and provides the maximum amount of clinical detail available. While contemporary clinical isolates lack the depth of phenotypic characterisation of well-studied laboratory strains, we believe their greater physiological relevance justifies this trade-off. Additionally, we are cautious about the potential for misdirecting readers or prompting overinterpretation by giving undue prominence to phenotypic details of the bacteria used that may or may not be relevant: a point already conceded above regarding the acapsular nature of USA300.

They also do not offer any evidence of the role of virulence factors for these strains as they did for SP. For example, do we know anything about the capsules of either EC or KP?

Author Response: The non-*S. pyogenes* bacterial strains assessed in the paper were selected based on their clinical association with bacteraemia, rather than their production of capsule/extracellular polysaccharides or other virulence factors. The hypothesis we set out in the Discussion, that the physical properties bestowed by capsule/extracellular polysaccharides may be a prominent driver of lymphatic metastasis, is included as a basis for future exploration, rather than a mechanistic focus of the paper. Therefore, the virulence factors - which are not yet fully characterised in these clinical isolates - are not explored in the manuscript.

However, the identities of such virulence factors are currently under investigation.

[Redacted]

[Redacted]

A comprehensive mechanistic study of the numerous bacterial species and strains that we have evaluated, conducted to the same high standards as undertaken for *S. pyogenes*, is a significant undertaking that itself would likely span several years work. We believe that attempting to amalgamate additional data concerning different virulence factors from other bacterial species would compromise understanding of the current manuscript, which is already data-rich and contains exciting and impactful findings. We hope that after reviewing our revisions that you will also share this opinion. We anticipate and hope that increased awareness of the extent of extracellular bacterial lymphatic metastasis in our model will lead other groups to consider and explore the phenomenon and consequences in their own bacterial infection and immunity models.

(Pseudomonas does not have a capsular polysaccharide)

Author Response: Thank you for highlighting this important distinction: that the alginate, Pel and Psl production by *P. aeruginosa* does not constitute a proper capsule. Accordingly, we have amended the text to include reference to 'extracellular polysaccharides' (EPS). Notably, hyaluronan is not covalently linked to the *S. pyogenes* cell surface either. Yet, as it is invariably referred to as 'hyaluronan/hyaluronic acid/hyaluronate capsule' in the literature, we will maintain convention to prevent confusion.

Does the anti-lymphatic endothelial receptor – (LYVE-1) also play a role for these other bacteria?

Author Response: We have not conducted experiments using anti-LYVE-1 antibodies in the newer infection models employed with other species of bacteria. However, as LYVE-1 is the main receptor for hyaluronan in the lymphatics and its binding specificity for this glycosaminoglycan is absolute (even the structurally related chondroitin, dermatan and heparan polymers are not recognised) and strictly dependent on avidity⁸, other extracellular polysaccharides of distinct molecular structure would not be expected to bind LYVE-1.

Having shown their technical expertise in delineating the lymphatics and isolating cellular elements within the lymphatics, it would have been quite illuminating for them to examine the lymphatic fluid. For example, is there immunoglobulin within the lymphatics??

Author Response: Thank you for this interesting suggestion. Though we share your opinion that analysis of the lymph could be valuable, we have not been able to achieve this experimentally for the flank lymphatics. During an immune response, immunoglobulin has been shown to be present in efferent lymph collected from the thoracic duct⁹ and Ig in lymph could feasibly influence phagocytosis and infection of bacteria transiting within lymphatic vessels. However, in our model of

acute bacterial infection, using specific pathogen free mice (which have not been prior exposed to *S. pyogenes*) and early ≤ 24 -hour time points, there would not be sufficient time to mount an antibody response and hence immunoglobulins (other than natural antibodies) would not be expected to play a major role. It is an important question however, in humans, where prior immunity may be an important factor, and where intravenous immunoglobulin is sometimes administered to combat invasive *S. pyogenes* infection.

In their discussion they state (p. 16, l.25) that the lymphatics do not “routinely allow every trivial bacterial infection” to develop into a serious systemic disease. Might they speculate on what types of protective mechanisms may come to bear?

Author Response: Our work strongly suggests an important role for neutrophils. This is most clearly illustrated by the transformation of avirulent *L. lactis* into a bacterium capable of lymphatic and systemic spread upon heterologous expression of the chemokine-cleaving protein, SpyCEP, which impedes neutrophil recruitment (Extended Data Fig. 5i). Human neutrophils, unlike murine neutrophils, produce antimicrobial defensins^{10,11}, and any effect of neutrophils in preventing lymphatic spread and development of systemic disease in mice may be even stronger in humans.

Speaking more broadly, analysis of invasive streptococcal disease indicates that those with chronic disease and skin lesions are more at risk^{12,13}, similarly host genetic factors^{14,15} and prior blunt force trauma^{16,17} are implicated. We would speculate that even in health, a low level of sporadic bacterial transit to lymph nodes might occur but would quickly be controlled by the host. However, under certain conditions, this balance might swing towards bacterial replication: i.e. a particularly virulent strain¹⁸ able to spread in greater numbers and more quickly due to increased lymphatic flow and access to the lymphatics following tissue injury, in a host predisposed to severe infection through immune status or genetic factors. Of these genetic factors, HLA polymorphism has been suggested to influence the susceptibility to severe *S. pyogenes* disease^{14,15}. In other streptococcal bacterial invasive disease, genetic variation in *NFKBIL2* which encodes the proinflammatory transcription factor NF- κ B inhibitor I κ B-R has been implemented¹⁹.

If SP secrete superantigens, why is there not a greater immune response within the LN given the large number of T cells and antigen-presenting cells?? Certainly if the authors suggest that the persistent stimulus of superantigen-secreting SP might induce autoimmune disease, there should be some evidence of immune response (e.g. germinal center formation).

Author Response: Streptococcal superantigens are highly specific to human MHC II (HLA) and so do not mediate notable T cell mitogenicity or other immune effects in non-humanised mice such as

those used in the current paper²⁰. As this study analyses early timepoints in acute bacterial disease, we do not believe this limitation is of significance. It may represent an important barrier to subsequent investigations into later time points (days to weeks) and longer study of the effects of lymphatic spread may necessitate the use of HLA class II transgenic mice. The authors have previously shown a marked expansion of draining lymph nodes in HLA II transgenic mice that was specific to the superantigen SpeA; at the time, bacterial transit to the draining lymph node had not been expected²¹.

Specific Comments: Given my comments above, the title should be changed to substitute “*S. pyogenes*” in place of “extracellular bacteria”. They should also limit the conclusion in the abstract to SP and not “pathogenic extracellular bacteria”, and new perspectives on SP pathogenesis rather than “bacterial” pathogenesis.

Author Response: In response to Reviewer 3’s earlier suggestion we have undertaken a significant number of additional in vivo experiments with multiple clinical strains representing six non-*S. pyogenes* species, consisting of both Gram positive and negative bacteria. The majority of these bacteria exhibited the behaviour of lymphatic-dominated spread and the data point to a generalised phenomenon, rather than something limited to *S. pyogenes*. Therefore, we feel the use of ‘extracellular bacteria’ is a fairer representation of the data and it might seem misleading or disingenuous to suggest the phenomenon was restricted to *S. pyogenes*. Furthermore, the terminology of the title is consistent with Journal convention. and will help the data reach scientists who study non-*S. pyogenes* bacteria, who we believe would be interested in the results of our work and its application to other bacteria.

Figure 2, panel “1”: to which cell surface are they referring?

Author Response: Apologies for this lack of clarity. The word leukocyte has now been added to the legend of Fig. 2: ‘fluorescently labelled streptococci adhered to **leukocyte** cell surface’

p.8, Fig 3: Why do the SP not enter the LN parenchyma?

Author Response: Passage of molecules through the floor of subcapsular sinus-lining endothelium into parenchymal conduits is physically restricted by a diaphragm comprised of the plasmalemma vesicle-associated protein (PLVAP) that covers transendothelial channels and deflects molecules in excess of around 70 kDa²².

They should explain why they want to identify podoplanin—are they citing it as a specific lymphatic vessel marker or as a link between cardiovascular and lymphatic systems?

Author Response: Podoplanin was employed to stain lymphatic endothelium to highlight lymph node sinuses. An explanation has been added to all relevant figure legends and in the text where relevant: ‘Podoplanin staining highlights lymphatic endothelial cells in the subcapsular and medullary sinuses, as well as fibroblastic reticular cells in the cortex of the lymph node’.

p. 9. Interestingly, the hyaluronan capsule’s role in virulence appears to be not so much in evading phagocytes but in retaining SP within the lymphatics based on their interaction with LYVE-1—is that correct? If so, this would be different from the role of capsule in *Klebsiella* and *E. coli*.

Author Response: Our data indicates that hyaluronan capsule plays three roles that contribute to lymphatic dissemination. Firstly, evasion of uptake by phagocytes (*Fig. 5, Fig. 6, Extended Data Fig. 5a, h, Extended Fig. 6c,d*); Secondly, promotion of retention within the lymphatics through specific interaction with LYVE-1¹ (*Extended Data Fig. 4d, e*); and thirdly, potential exploitation of its physical properties of hydrophobicity and polyanionic nature to enhance lymphatic uptake (*Extended Data Fig. 4e*). Of these roles, only the first – specific interaction with LYVE-1 – would be expected to be unique to *S. pyogenes* and not expected with *Klebsiella* or *E. coli*. Our hypothesis is that capsule/EPS might enhance lymphatic spread and drive systemic disease in other species of bacteria, like *Klebsiella* through alteration of physical properties that lead to enhanced lymphatic uptake.

Is there any evidence of germinal center formation within the lymph nodes transited by the SP?

Author Response: While this is an interesting suggestion for later timepoints, in the current manuscript we focus on early timepoints (<24 hours) in acute infection. Hence, we have not analysed for germinal centres, which typically form after this time point. Overall, it is unlikely that adaptive immunity contributes to the pathogen response within 24 hours in previously unexposed mice. We are unable to analyse later timepoints with the current model of acute infection, as ethical endpoints are reached shortly after 24 hours. Though alterations to achieve this are a focus of future investigations.

How did the SP leave the lymphatics at 24 hr and enter the LN parenchyma??

Author Response: Characteristic smeared DAPI staining 24 hours post infection, that is not present at early timepoints, indicates widespread necrosis around concentrated populations of bacteria (*Fig. 5e, f and Extended Data Fig. 5b, c, e, f*). Death of host cells and disruption of endothelial barriers may allow bacterial spread deeper into the lymph node. Damage to cells is likely mediated by bacterial

cytotoxicity²³, though host-mediated damage from cellular responses and inflammation could also contribute²⁴. Furthermore, there are many bacteria in the blood at 24 hours post infection (*Fig. 1c*) and these bacteria could be visible in the blood vessels running throughout the lymph node parenchyma, possibly also contributing to invasion.

p.11. Was there any effect of SpyCEP mutants on PMN recruitment to the original hindlimb site of infection??

Author Response: This was not directly tested in our study, but we and others^{25,26}, have previously demonstrated that SpyCEP reduces, but does not abrogate neutrophil recruitment to bacteria at an original site of infection. The impact on bacterial burden at the site of infection is modest, if detectable, compared with the impact on lymph node and bloodstream spread. We have now added additional experimental data to the manuscript concerning the role of SpyCEP. We use heterologously expressed SpyCEP in *Lactococcus lactis* (an avirulent bacterium that is readily phagocytosed, lacks virulence factors and does not naturally express SpyCEP), to determine the contribution of SpyCEP to dissemination without masking of the effects resulting from the other many virulence factors of pathogenic bacteria like *S. pyogenes*. When *L. lactis* with an empty control plasmid was injected into mice, the bacteria were confined to the injection site and did not disseminate, yet rather showed a 1.5 log fold reduction in CFU. In clear contrast, *L. lactis* expressing SpyCEP showed striking dissemination to draining lymph nodes and systemic organs (*Extended Data Fig. 5i and discussed: Page 12; lines 17–21*). This data is consistent with previous reports and supports an action of neutrophils in clearing *L. lactis* from both the original site of infection, and particularly metastatic infection sites.

p. 20., Methods. It appears that the bacteria were grown overnight before use and hence in stationary phase. Was there any comparison of bacteria grown to log versus stationary phase??

Author Response: We thank the Reviewer for raising this potentially important issue. In response to their concern, we have now undertaken a comparison of bacteria grown to log versus stationary phase (*Extended Data Fig. 2b*). Lymphatic spread was shown to occur with both stationary and log phase bacteria.

Mice: In their methods they claim to use FVB/n, BALB/c and C57BL/6 mice, yet nowhere in their figure legends do they indicate the strain of mice. And why the three different strains??

Author Response: We apologise for this careless omission. The strain of mice used is now stated in each relevant figure legend. FVB/n mice were used unless otherwise stated, consistent with our

previous work¹. BALB/c mice were used for data presented in *Extended Data Fig 2a*, to confirm that metastasis occurred in another common strain of mice. Finally, C57BL/6 mice were used in some pilot studies and *Extended Data Fig. 5i*, based on availability. Ultimately, *S. pyogenes* exhibited lymphatic spread in these three strains of mice.

References

- 1 Lyskey, N. N. *et al.* Rapid Lymphatic Dissemination of Encapsulated Group A Streptococci via Lymphatic Vessel Endothelial Receptor-1 Interaction. *PLoS Pathog* **11**, e1005137, doi:10.1371/journal.ppat.1005137 (2015).
- 2 Vincent, P. *et al.* Bacterial translocation in patients with colorectal cancer. *J Infect Dis* **158**, 1395-1396, doi:10.1093/infdis/158.6.1395 (1988).
- 3 Sakamoto, H. *et al.* Isolation of bacteria from cervical lymph nodes in patients with oral cancer. *Arch Oral Biol* **44**, 789-793, doi:10.1016/s0003-9969(99)00079-5 (1999).
- 4 O'Boyle, C. J. *et al.* Microbiology of bacterial translocation in humans. *Gut* **42**, 29-35, doi:10.1136/gut.42.1.29 (1998).
- 5 Wells, C. L., Jechorek, R. P., Twiggs, L. B. & Brooker, D. C. Recovery of viable bacteria from pelvic lymph nodes of patients with gynecologic tumors. *J Infect Dis* **162**, 1216-1218, doi:10.1093/infdis/162.5.1216-a (1990).
- 6 Sakamoto, H., Sasaki, J. & Nord, C. E. Association between bacterial colonization on the tumor, bacterial translocation to the cervical lymph nodes and subsequent postoperative infection in patients with oral cancer. *Clin Microbiol Infect* **5**, 612-616, doi:10.1111/j.1469-0691.1999.tb00417.x (1999).
- 7 Shallcross, L. J. *et al.* Cohort study protocol: Bioresource in Adult Infectious Diseases (BioAID). *Wellcome Open Res* **3**, 97, doi:10.12688/wellcomeopenres.14690.1 (2018).
- 8 Lawrance, W., Banerji, S., Day, A. J., Bhattacharjee, S. & Jackson, D. G. Binding of Hyaluronan to the Native Lymphatic Vessel Endothelial Receptor LYVE-1 Is Critically Dependent on Receptor Clustering and Hyaluronan Organization. *J Biol Chem* **291**, 8014-8030, doi:10.1074/jbc.M115.708305 (2016).
- 9 Haig, D. M., Hopkins, J. & Miller, H. R. Local immune responses in afferent and efferent lymph. *Immunology* **96**, 155-163 (1999).
- 10 Eisenhauer, P. B. & Lehrer, R. I. Mouse neutrophils lack defensins. *Infect Immun* **60**, 3446-3447 (1992).
- 11 Risso, A. Leukocyte antimicrobial peptides: multifunctional effector molecules of innate immunity. *J Leukoc Biol* **68**, 785-792 (2000).
- 12 Factor, S. H. *et al.* Risk factors for pediatric invasive group A streptococcal disease. *Emerg Infect Dis* **11**, 1062-1066, doi:10.3201/eid1107.040900 (2005).
- 13 Factor, S. H. *et al.* Invasive group A streptococcal disease: risk factors for adults. *Emerg Infect Dis* **9**, 970-977, doi:10.3201/eid0908.020745 (2003).
- 14 Parks, T. *et al.* Elevated risk of invasive group A streptococcal disease and host genetic variation in the human leucocyte antigen locus. *Genes Immun*, doi:10.1038/s41435-019-0082-z (2019).

- 15 Kotb, M. *et al.* An immunogenetic and molecular basis for differences in outcomes of invasive group A streptococcal infections. *Nat Med* **8**, 1398-1404, doi:10.1038/nm800 (2002).
- 16 Nuwayhid, Z. B., Aronoff, D. M. & Mulla, Z. D. Blunt trauma as a risk factor for group A streptococcal necrotizing fasciitis. *Ann Epidemiol* **17**, 878-881, doi:10.1016/j.annepidem.2007.05.011 (2007).
- 17 Lamb, L. E. *et al.* Impact of contusion injury on intramuscular emm1 group a streptococcus infection and lymphatic spread. *Virulence* **9**, 1074-1084, doi:10.1080/21505594.2018.1482180 (2018).
- 18 Cole, J. N. *et al.* Trigger for group A streptococcal M1T1 invasive disease. *FASEB J* **20**, 1745-1747, doi:10.1096/fj.06-5804fje (2006).
- 19 Chapman, S. J. *et al.* Common NFKBIL2 polymorphisms and susceptibility to pneumococcal disease: a genetic association study. *Crit Care* **14**, R227, doi:10.1186/cc9377 (2010).
- 20 Sriskandan, S. *et al.* Enhanced susceptibility to superantigen-associated streptococcal sepsis in human leukocyte antigen-DQ transgenic mice. *J Infect Dis* **184**, 166-173, doi:10.1086/322018 (2001).
- 21 Sriskandan, S. *et al.* Streptococcal pyrogenic exotoxin A release, distribution, and role in a murine model of fasciitis and multiorgan failure due to *Streptococcus pyogenes*. *J Infect Dis* **173**, 1399-1407 (1996).
- 22 Rantakari, P. *et al.* The endothelial protein PLVAP in lymphatics controls the entry of lymphocytes and antigens into lymph nodes. *Nat Immunol* **16**, 386-396, doi:10.1038/ni.3101 (2015).
- 23 Sierig, G., Cywes, C., Wessels, M. R. & Ashbaugh, C. D. Cytotoxic effects of streptolysin o and streptolysin s enhance the virulence of poorly encapsulated group a streptococci. *Infect Immun* **71**, 446-455, doi:10.1128/iai.71.1.446-455.2003 (2003).
- 24 Schwager, S. & Detmar, M. Inflammation and Lymphatic Function. *Front Immunol* **10**, 308, doi:10.3389/fimmu.2019.00308 (2019).
- 25 Zinkernagel, A. S. *et al.* The IL-8 protease SpyCEP/ScpC of group A *Streptococcus* promotes resistance to neutrophil killing. *Cell Host Microbe* **4**, 170-178, doi:10.1016/j.chom.2008.07.002 (2008).
- 26 Kurupati, P. *et al.* Chemokine-cleaving *Streptococcus pyogenes* protease SpyCEP is necessary and sufficient for bacterial dissemination within soft tissues and the respiratory tract. *Mol Microbiol* **76**, 1387-1397, doi:10.1111/j.1365-2958.2010.07065.x (2010).

REVIEWER COMMENTS

Reviewer #1 (Remarks to the Author):

The authors present data showing that multiple bacteria remain extracellular while disseminating from a local site of infection through multiple sequential draining lymph nodes in order to reach the bloodstream via efferent postnodal lymphatic vessels. These data provide new insight and a new mechanism of systemic bacterial dissemination that does not rely on phagocytosis and intracellular transport by macrophages or neutrophils. The experiments are well-performed and the data support the main conclusions of the manuscript. The revised manuscript provides further mechanistic support and is more clearly written. I have only annoying, nitpicking comments that can be ignored, but my OCD requires me to document them here.

Comments:

Abstract Line 20: For the lymphatic system, conduits refer to the collagen fiber structures traversing lymph nodes. In the context of this sentence, please use “lymphatic vessels” or “initial lymphatic vessels” instead of “lymphatic conduits”.

Page 19: Lines 19-22: “Hence, it is feasible that bacterial lymphatic metastasis may play a prominent role in the slow development of natural protective immunity and common recurrent childhood infections observed with *S. pyogenes* and other bacteria”. I have trouble following this sentence. Should “immunity and common” be immunity against common? Or are the authors suggesting that lymphatic metastasis play a prominent role in common recurrent childhood infections? Also, I don’t follow the “Hence”. The preceding sentences would make it seem that no immune response should form. This section should be clarified.

Page 20: Line 8; Missing “did” between “bacteria” and “not”.

Reviewer #2 (Remarks to the Author):

authors have addressed convincingly all issues raised during revision.

Reviewer #3 (Remarks to the Author):

The authors have done an excellent job of responding to the reviews. However, I still have a remaining concern. In response to my earlier review stating that the title should focus on *S. pyogenes* (SP) and not "extracellular bacteria", the authors added additional extracellular bacteria to justify their title, but not without raising new concerns. They spend considerable time correlating SP virulence factors with the lymphatic spread, and then show that with isolates of *Klebsiella* (KP), *Pseudomonas aeruginosa* (PA) and *E. coli* (EC) they also detect sequential lymphatic spread resulting in disseminated infection. Unlike the case with SP, they do not identify any virulence factors with these isolates. Moreover, there are very disparate results within each of their strains (Figure 7). For example, the dissemination differs between the two KP, PA and EC strains, and the CFU levels differ considerably. Is this due to virulence factors? These data are unlike those shown with multiple SP strains. How are these differences explained? Without doing a detailed analysis of the respective virulence factors the authors might report what the LD50s are of the various strains to see if the level of spread correlates with the LD50. They may find, for example, the EC ST131 strain is not very virulent in mice.

In my comments about examining lymphatic fluid for immune constituents, I was not so much interested in looking for evidence of an adaptive immune response, particularly since they were looking very early in the immune response, but wondering if the lymphatic fluid had natural antibody, complement etc.

Author Response to Peer Review: (NComm19-38830A)

Siggins et al: Lymphatic Metastasis of Virulent Extracellular Bacteria Drives Systemic Infection

We thank all of the Reviewers for their supportive and insightful comments in relation to the Revision. Point by point responses and details of changes follow below. Please note that as a consequence of discussion with the Editor, the paper now only includes data concerning *Streptococcus pyogenes*. As such, the manuscript has been retitled '*Extracellular bacterial lymphatic metastasis drives Streptococcus pyogenes systemic infection*' and references to other bacterial species removed, bar a brief generic section in the Discussion.

Reviewer #1 (Remarks to the Author):

The authors present data showing that multiple bacteria remain extracellular while disseminating from a local site of infection through multiple sequential draining lymph nodes in order to reach the bloodstream via efferent postnodal lymphatic vessels. These data provide new insight and a new mechanism of systemic bacterial dissemination that does not rely on phagocytosis and intracellular transport by macrophages or neutrophils. The experiments are well-performed and the data support the main conclusions of the manuscript. The revised manuscript provides further mechanistic support and is more clearly written. I have only annoying, nitpicking comments that can be ignored, but my OCD requires me to document them here.

Author response: Thank you for your praise of our work and your helpful input. Responses to your individual points are given below.

Comments:

Abstract Line 20: For the lymphatic system, conduits refer to the collagen fiber structures traversing lymph nodes. In the context of this sentence, please use "lymphatic vessels" or "initial lymphatic vessels" instead of "lymphatic conduits".

Author response: Thank you for highlighting this. The phrasing '*lymphatic vessels*' is now used instead.

Page 19: Lines 19-22: "Hence, it is feasible that bacterial lymphatic metastasis may play a prominent role in the slow development of natural protective immunity and common recurrent childhood infections observed with S. pyogenes and other bacteria". I have trouble following this sentence. Should "immunity and common" by immunity against common? Or are the authors suggesting that lymphatic metastasis play a prominent role in common recurrent childhood infections? Also, I don't follow the "Hence". The preceding sentences would make it seem that no immune response should form. This section should be clarified.

Author response: We apologise for the lack of clarity. We have replaced the previous text with the sentence: '*Bacterial lymphatic metastasis may contribute to recurrent childhood streptococcal infections⁵⁷ by slowing natural development of protective immunity*'.

Page 20: Line 8; Missing "did" between "bacteria" and "not".

Author response: We apologise for this error which is now corrected.

Reviewer #2 (Remarks to the Author):

authors have addressed convincingly all issues raised during revision.

Author response: Thank you for your time and valuable input.

Reviewer #3 (Remarks to the Author):

The authors have done an excellent job of responding to the reviews.

Author response: Thank you. Further responses to remaining concerns are given below.

*However, I still have a remaining concern. In response to my earlier review stating that the title should focus on *S. pyogenes* (SP) and not "extracellular bacteria", the authors added additional extracellular bacteria to justify their title, but not without raising new concerns. They spend considerable time correlating SP virulence factors with the lymphatic spread, and then show that with isolates of *Klebsiella* (KP), *Pseudomonas aeruginosa* (PA) and *E. coli* (EC) they also detect sequential lymphatic spread resulting in disseminated infection. Unlike the case with SP, they do not identify any virulence factors with these isolates. Moreover, there are very disparate results within each of their strains (Figure 7). For example, the dissemination differs between the two KP, PA and EC strains, and the CFU levels differ considerably. Is this due to virulence factors? These data are unlike those shown with multiple SP strains. How are these differences explained? Without doing a detailed analysis of the respective virulence factors the authors might report what the LD50s are of the various strains to see if the level of spread correlates with the LD50. They may find, for example, the EC ST131 strain is not very virulent in mice.*

Author response: Thank you for raising these valid points. The purpose of extending the experiments to a greater number of bacterial species (and representative strains) was to confirm that lymphatic spread could be observed among other extracellular bacterial species, in part because USA300 might not have been ideal, but also in response to strong Editorial recommendation. We were not trying to address intra-species variation but simply exploring if the phenomenon was also observed for other species, which it was.

We agree that the differences observed between strains are likely due to the relative ability to clear each strain (i.e. strain virulence), as development of bacteraemia is a good indicator of relative virulence in mice. Assessment of LD₅₀ is not within the scope of our licenses, and we believe that a more quantitative metric such as bacterial clearance would reproduce the variation already observed between strains. Nonetheless, the other bacteria did appear to use the lymphatic system-which was the point of extending the experiments to other species, rather than to understand the wider question of differences in virulence or the mechanisms that might allow other bacteria to reach lymph nodes.

We indicated to the Editor that we were willing to adjust the content of the paper and title as requested and have now done so. The manuscript has been retitled 'Extracellular bacterial lymphatic metastasis drives *Streptococcus pyogenes* systemic infection' and references to data from other bacterial species have been removed, bar a brief section in the Discussion. We plan to explore the lymphatic metastasis of other bacterial species in further depth in future work.

In my comments about examining lymphatic fluid for immune constituents, I was not so much interested in looking for evidence of an adaptive immune response, particularly since they were looking very early in the immune response, but wondering if the lymphatic fluid had natural antibody, complement etc.

Author response: Thank you for further clarifying your query. We are unable to sample murine flank lymphatics to provide a direct experimental answer to the question. There is limited literature detailing lymph concentrations of natural antibodies and complement. However, complement has been reported to be present in low-levels in lymph¹. Natural antibodies are produced by spleen resident B1a lymphocytes and by B1 lymphocytes of bone marrow. As B-1 cells are present at relatively low levels in lymph nodes, we predict that the concentration of natural antibodies in lymph would be even lower than the low levels measured in plasma². Despite this, it is entirely plausible that natural antibodies and complement within lymph could contribute to antibacterial defence, and this is an interesting area for further study.

References

- 1 Olszewski, W. L. & Engeset, A. Haemolytic complement in peripheral lymph of normal men. *Clin Exp Immunol* **32**, 392-398 (1978).
- 2 Palma, J., Tokarz-Deptula, B., Deptula, J. & Deptula, W. Natural antibodies - facts known and unknown. *Cent Eur J Immunol* **43**, 466-475, doi:10.5114/ceji.2018.81354 (2018).